# GaussianAnything: Interactive Point Cloud Flow Matching for 3D Object Generation

**Yushi Lan**[†]**, Shangchen Zhou**[†]**, Zhaoyang Lyu**[α]**, Fangzhou Hong**[†]**,**
**Shuai Yang**[β]**, Bo Dai**[γ]**, Xingang Pan**[†]**, Chen Change Loy**[†]
[†]S-Lab, Nanyang Technological University, Singapore
[α]Shanghai Artificial Intelligence Laboratory, [β]WICT, Peking University
[γ]The University of Hong Kong
https://nirvanalan.github.io/projects/GA/

## Abstract

While 3D content generation has advanced significantly, existing methods still face challenges with input formats, latent space design, and output representations. This paper introduces a novel 3D generation framework that addresses these challenges, offering scalable, high-quality 3D generation with an interactive *Point Cloud-structured Latent* space. Our framework employs a Variational Autoencoder (VAE) with multi-view posed RGB-D(epth)-N(ormal) renderings as input, using a unique latent space design that preserves 3D shape information, and incorporates a cascaded latent flow-based model for improved shape-texture disentanglement. The proposed method, GaussianAnything, supports multi-modal conditional 3D generation, allowing for point cloud, caption, and single image inputs. Notably, the newly proposed latent space naturally enables geometry-texture disentanglement, thus allowing 3D-aware editing. Experimental results demonstrate the effectiveness of our approach on multiple datasets, outperforming existing native 3D methods in both text- and image-conditioned 3D generation.

## 1 Introduction

3D content generation holds great potential for transforming the virtual reality, film, and gaming industries. Current approaches typically follow one of two paths: either a 2D-lifting method or the design of native 3D diffusion models. While the 2D-lifting approach (Shi et al., 2023b; Liu et al., 2023b) benefits from leveraging 2D diffusion model priors, it is often hindered by expensive optimization, the Janus problem, and inconsistencies between views. In contrast, native 3D diffusion models (Jun & Nichol, 2023; Lan et al., 2024a; Zhang et al., 2024b) are trained from scratch for 3D generation, offering improved generality, efficiency, and control.

Despite the progress in native 3D diffusion models, several design challenges still persist: **(1)** *Input format to the 3D VAE.* Most methods (Zhang et al., 2024b; Li et al., 2024) directly adopt point cloud as input. However, it fails to encode the high-frequency details from textures. Besides, this limits the available training dataset to artist-created 3D assets, which are challenging to collect on a large scale. LN3Diff (Lan et al., 2024a) adopt multi-view images as input. Though straightforward, it lacks direct 3D information input and cannot comprehensively encode the given object. **(2)** *3D latent space structure.* Since 3D objects are diverse in geometry, color, and size, most 3D VAE models adopt the permutation-invariant *set latent* (Zhang et al., 2023; Sajjadi et al., 2022; Zhang et al., 2024b) to encode incoming 3D objects. Though flexible, this design lacks the image-latent correspondence as in Stable Diffusion VAE (Rombach et al., 2022), where the VAE latent code can directly serve as the proxy for editing input image (Mou et al., 2023b;a). Other methods adopt latent tri-plane (Wu et al., 2024; Lan et al., 2024a) as the 3D latent representation. However, the latent tri-plane is still unsuitable for interactive 3D editing as changes in one plane may not map to the exact part of the objects that need editing. **(3)** *Choice of 3D output representations.* Existing solutions either output texture-less SDF (Wu et al., 2024; Zhang et al., 2024b), which requires additional shading model for post-processing; or volumetric tri-plane (Lan et al., 2024a), which struggles with high-resolution rendering due to extensive memory required by volumetric rendering (Mildenhall et al., 2020).

In this study, we propose a novel 3D generation framework that resolves the problems above and enables scalable, high-quality 3D generation with an interactive *Point Cloud-structured Latent* space. The resulting method, dubbed GAUSSIANANYTHING, supports multi-modal conditional 3D generation, including point cloud, caption, and image. Specifically, we propose a 3D VAE that adopts multi-view posed RGB-D(epth)-N(ormal) renderings as the input, which are easy to render and contain comprehensive 3D attributes corresponding to the input 3D object. The information of each input view is channel-wise concatenated and efficiently encoded with the scene representation transformer (Sajjadi et al., 2022), yielding a *set latent* (Zhang et al., 2023) that compactly encodes the given 3D input. Instead of directly applying it for diffusion learning (Zhang et al., 2024b; Li et al., 2024), our novel design concretizes the unordered tokens into the shape of the 3D input. Specifically, this is achieved by cross-attending (Huang et al., 2024b) the *set latent* via a sparse point cloud sampled from the input 3D shape, as visualized in Fig. 1. The resulting point-cloud structured latent space significantly facilitate shape-texture disentanglement and 3D editing. Afterward, a DiT-based 3D decoder (Peebles & Xie, 2023; Lan et al., 2024a) gradually decodes and upsamples the latent point cloud into a set of dense surfel Gaussians (Huang et al., 2024a), which are rasterized to high-resolution renderings to supervise 3D VAE training.

After the 3D VAE is trained, we conduct cascaded latent diffusion modeling on the latent space through flow matching (Albergo et al., 2023; Lipman et al., 2023; Liu et al., 2023c) using the DiT (Peebles & Xie, 2023) framework. To encourage better shape-texture disentanglement, a point cloud diffusion model is first trained to carve the overall layout of the input shape. Then, a point-cloud feature diffusion model is cascaded to output the corresponding feature conditioned on the generated point cloud. The generated featured point cloud is then decoded into surfel Gaussians via pre-trained VAE for downstream applications.

In summary, we contribute a comprehensive 3D generation framework with a point cloud-structured 3D latent space. The redesigned 3D VAE efficiently encodes the 3D input into an interactive latent space, which is further decoded into high-quality surfel Gaussians. The diffusion models trained on the compressed latent space have shown superior performance in text-conditioned 3D generation and editing, as well as impressive image-conditioned 3D generation on general real world data.

## 2 RELATED WORK

**3D Generation via 2D Diffusion Models.** The success of 2D diffusion models (Song et al., 2021; Ho et al., 2020) has inspired their application to 3D generation. Score distillation sampling (Poole et al., 2022; Wang et al., 2023) distills 3D from a 2D diffusion model, but faces challenges like expensive optimization, mode collapse, and the Janus problem. More recent methods propose learning the 3D via a two-stage pipeline: multi-view images generation (Shi et al., 2023b; Long et al., 2024; Shi et al., 2023a) and feed-forward 3D reconstruction and editing (Hong et al., 2024b; Xu et al., 2024a; Tang et al., 2024; Xu et al., 2024b; Chen et al., 2024b). However, their performance is bounded by the multi-view generation results, which usually violate view consistency (Liu et al., 2023b) and fail to scale up to higher resolution (Shi et al., 2023a). Moreover, this two-stage pipeline limits the 3D editing capability due to the lack of a 3D-aware latent space.

**Native 3D Diffusion Models.** Native 3D diffusion models (Zhang et al., 2023; Zeng et al., 2022; Zhang et al., 2024b; Lan et al., 2024a; Li et al., 2024) are recently proposed to achieve high-quality, efficient and scalable 3D generation. A native 3D diffusion pipeline involves a two-stage training process: encoding 3D objects into the VAE latent space (Kingma & Welling, 2013; Kosiorek et al., 2021), and latent diffusion model on the corresponding latent codes. Though straightforward, existing methods differ in VAE input formats, latent space structure and output 3D representations. While most methods adopt point alone as the VAE input (Zhang et al., 2023; 2024b; Li et al., 2024), our proposed method encodes a hybrid 3D information through convolutional encoder. Moreover, comparing to the latent set (Zhang et al., 2023; Sajjadi et al., 2022) representation, our proposed method adopts a point cloud-structured latent space, which can be directly used for interactive 3D editing. Besides, rather than producing textureless SDF, our method directly decodes the 3D latent codes into high-quality surfel Gaussians (Huang et al., 2024a), which can be directly used for efficient rendering.

**Point-based Shape Representation and Rendering.** The proliferation of 3D scanners and RGB-D cameras makes the capture and processing of 3D point clouds commonplace (Gross & Pfister,

2011). In the era of deep learning, learning-based methods are emerging for point set processing (Qi et al., 2016; Zhao et al., 2021), up-sampling (Yu et al., 2018), shape representation (Genova et al., 2020; Lan et al., 2024b), and rendering (Pfister et al., 2000; Yifan et al., 2019; Lassner & Zollhöfer, 2021; Xu et al., 2022; Kerbl et al., 2023). Moreover, given its affinity for modern network architectures (Huang et al., 2024b; Zhao et al., 2021), more explicit nature against other 3D representations (Chan et al., 2022; Mildenhall et al., 2020; Müller et al., 2022), efficient rendering (Kerbl et al., 2023), and even high-quality surface modeling (Huang et al., 2024a), point-based 3D representations are rapidly developing towards the canonical 3D representation for learning 3D shapes. Thus, we choose (featured) point cloud as the representation for the 3D VAE latent space, and 2D Gaussians (Huang et al., 2024a) as the output 3D representations.

**Feed-forward 3D Reconstruction and View Synthesis.** To bypass the per-scene optimization of NeRF, researchers have proposed learning a prior model through image-based rendering (Wang et al., 2021; Yu et al., 2021). However, these methods are primarily designed for view synthesis and lack explicit 3D representations. Sajjadi et al. (2022; 2023) propose Scene representation transformer (SRT) to process RGB images with Vision Transformer (Dosovitskiy et al., 2021) and infers a *set-latent scene representation*. Though benefiting from the flexible design, its geometry-free paradigm also fails to generate explicit 3D outputs. Recently, LRM-line of work (Hong et al., 2024b; Tang et al., 2024; Wang et al., 2024) have proposed a feed-forward framework for generalized monocular reconstruction. However, they are still regression-based models and lack the latent space designed for generative modeling and 3D editing. Besides, they are limited to 3D reconstruction only and fail to support other modalities.

## 3 GAUSSIANANYTHING

This section introduces our native 3D flow-based model, which learns 3D-aware flow-based prior over the novel point-cloud structured latent space through a dedicated 3D VAE. The goal of training is to learn

1. An encoder $\mathcal{E}_\phi$ that maps a set of posed RGB-D-N images $\mathcal{R} = \{R_i, ..., R_V\}$, corresponding to the given 3D object to a point-cloud structured latent $\mathbf{z} = [\mathbf{z}_x \oplus \mathbf{z}_h]$;

2. A conditional cascaded transformer denoiser $\boldsymbol{\epsilon}_\Theta^h(\mathbf{z}_{h,t}, \mathbf{z}_{x,0}, t, c) \circ \boldsymbol{\epsilon}_\Theta^x(\mathbf{z}_{x,t}, t, c)$ to denoise the noisy latent code $\mathbf{z}_t$ given the time step $t$ and condition prompt $c$;

3. A decoder $\mathcal{D}_\psi$ (including a Transformer $\mathcal{D}_T$ and a cascaded attention-base Upsampler $\mathcal{D}_U$) to map $\mathbf{z}_0$ to the surfel Gaussian $\widetilde{\mathcal{G}}$ corresponding to the input object. Moreover, our attention-based decoding of dense surfel Gaussian also provides a novel way for efficient Gaussian prediction

Beyond the advantages shared by existing 3D LDM (Zhang et al., 2024b; Lan et al., 2024a), our design offers a flexible point-cloud structured latent space and enables interactive 3D editing.

In the following subsections, we first discuss the proposed 3D VAE with a detailed framework design in Sec 3.1. Based on that, we introduce the cascaded conditional 3D flow-based model stage in Sec. 3.2. The method overview is shown in Fig. 1.

### 3.1 POINT-CLOUD STRUCTURED 3D VAE

Unlike image and video, the 3D domain is un-uniform and represented differently for different purposes. Thus, how to encode 3D objects into the latent space for flow-based model learning plays a crucial role in the 3D generation performance. This challenge is two-fold: what 3D representations to encode, and what network architecture to process the input.

**Versatile 3D Input.** Instead of using dense point cloud (Zhang et al., 2024b; Li et al., 2024), we adopt multi-view posed RGB-D(epth)-N(ormal) images as input, which encode the 3D input more comprehensively and can be efficiently processed by well-established network architectures (Sajjadi et al., 2022; Wu et al., 2023a) in a flexible manner. Specifically, the input is a set of multi-view renderings $\mathcal{R}$ of a 3D object, where each rendering within the set $R = (I, \Delta, N, \pi)$ contains thorough 3D attributes that depict the underlying 3D object from the given viewpoint: the rendered RGB

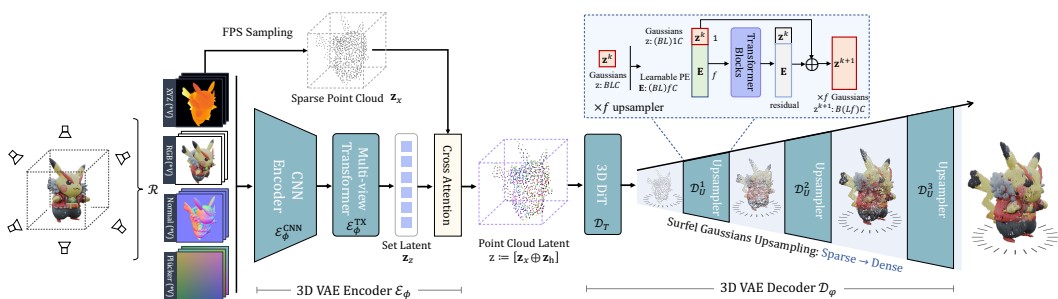

Figure 1: **Pipeline of the 3D VAE of GAUSSIANANYTHING.** In the 3D latent space learning stage, our proposed 3D VAE $\mathcal{E}_\phi$ encodes $V-$views of posed RGB-D(epth)-N(ormal) renderings $\mathcal{R}$ into a point-cloud structured latent space. This is achieved by first processing the multi-view inputs into the un-structured *set latent*, which is further projected onto the 3D manifold through a cross attention block, yielding the point-cloud structured latent code $\mathbf{z}$. The structured 3D latent is further decoded by a 3D-aware DiT transformer, giving the coarse Gaussian prediction. For high-quality rendering, the base Gaussian is further up-sampled by a series of cascaded upsampler $\mathcal{D}_U^k$ towards a dense Gaussian for high-resolution rasterization. The 3D VAE training objective is detailed in Eq. (5).

image $I \in \mathbb{R}^{H \times W \times 3}$, depth map $\Delta \in \mathbb{R}^{H \times W}$, normal map $N \in \mathbb{R}^{H \times W \times 3}$, and the corresponding camera pose $\pi$.

To unify these 3D attributes in the same format, we further process the camera $\pi$ into Plücker co-ordinates (Sitzmann et al., 2021) $\mathbf{p}_i = (\mathbf{o} \times \mathbf{d}_{u,v}, \mathbf{d}_{u,v}) \in \mathbb{R}^6$, where $\mathbf{o}_i \in \mathbb{R}^3$ is the camera origin, $\mathbf{d}_{u,v} \in \mathbb{R}^3$ is the normalized ray direction, and $\times$ denotes the cross product. Thus, the Plücker embedding of a given camera $\pi$ can be expressed as $\mathbf{P} \in \mathbb{R}^{H \times W \times 6}$. Besides, following MCC (Wu et al., 2023a), we use $\pi$ to unproject the depth map into their 3D positions $X \in \mathbb{R}^{H \times W \times 3}$. The resulting information is channel-wise concatenated, giving $\tilde{R} = [I \oplus X \oplus N \oplus \mathbf{P}] \in \mathbb{R}^{H \times W \times (3+3+3+6=15)}$.

**Transformer-based 3D Encoding.** Given the 3D renderings $\mathcal{R}$, encoding them into a 3D latent space remains a significant challenge. Independently processing each input rendering $\tilde{R}$ with existing network architecture (Wu et al., 2023a; Dosovitskiy et al., 2021) overlooks the information from other views, leading to 3D inconsistency and content drift across views (Liu et al., 2023b).

Existing multi-view generation alleviates this issue by injecting 3D attention (Shi et al., 2023b; Tang et al., 2024; Shi et al., 2023a) into the U-Net architecture. Inspired by its effectiveness, here we directly adopt Scene Representation Transformer (SRT)-like encoder (Sajjadi et al., 2022; 2023) to process the multi-view inputs, which fully adopts 3D attention transformer block for the 3D representation learning. Specifically, the encoder first down-samples the multi-view inputs via a shared CNN backbone, and then processes the aggregated multi-view tokens through the transformer encoder (Dosovitskiy et al., 2021):

$$\mathbf{z}_z = \mathcal{E}_\phi^{\text{TX}}(\mathcal{E}_\phi^{\text{CNN}}(\{\tilde{R}\})), \tag{1}$$

where $\mathbf{z}_z$ is the *set latent* corresponding to the 3D input. This can be seen as the full-attention version of the existing 3D attention-augmented architecture. The resulting latent codes $\mathbf{z}_z$ fully capture the intact 3D information corresponding to the input. Compared to existing work that adopts point clouds only as input (Zhang et al., 2024b; Li et al., 2024), our proposed solution supports more 3D properties as input in a flexible way. In addition, attention operations can be well optimized in modern GPU architecture (Dao et al., 2022; Dao, 2024).

**Point Cloud-structured Latent Space.** Though $\mathbf{z}_z$ fully captures the given 3D input, it is not ideal to serve as a latent space for our task due to the following limitations: 1) The latent space is cumbersome to perform flow matching. Specifically, $\mathbf{z}_z$ has a shape of $V \times (H/f) \times (W/f) \times C$, where $V$ is the number of input views, $H, W$ is the input resolution and $f$ is the down-sampling factor of the CNN backbone. Given $V = 8$, $f = 8$, and $H = W = 512$, the resulting latent codes will have the shape of $32768 \times C$. This latent space incurs a high computation cost for multi-view attention (Shi et al., 2023b). 2) The multi-view features $\mathbf{z}_z$ are not native 3D representations and naturally suffer from view inconsistency (Liu et al., 2023b) even with enough compute available (Shi et al., 2023a). 3) Since $\mathbf{z}_z$ is an un-structured *set* (Lee et al., 2019) 3D latent space (Zhang et al., 2023; 2024b), it also sacrifices an explicit, editable latent space (Mou et al., 2023a) for flexibility.

Here, we resolve these issues by proposing a point cloud-structured latent space. Specifically, we project the un-structured features $\mathbf{z}_z$ onto the sparse 3D point cloud of the input 3D shape through the cross attention layer:

$$\mathbf{z}_h := \text{CrossAttn}(\text{PE}(\mathbf{z}_x), \mathbf{z}_z, \mathbf{z}_z), \tag{2}$$

where $\text{CrossAttn}(Q, K, V)$ denotes a cross attention block with query $Q$, key $K$, and value $V$. $\mathbf{z}_x \in \mathbb{R}^{3 \times N}$ is a sparse point cloud sampled from the surface of the 3D input with Farthest Point Sampling (FPS) (Qi et al., 2017), and PE denotes positional embedding (Tancik et al., 2020). Intuitively, we define a *read* cross attention block (Huang et al., 2024b) that cross attends information from un-structured representation $\mathbf{z}_z$ into the point-cloud structured feature $\mathbf{z}_h \in \mathbb{R}^{C_h \times N}$, with $C_h \ll C$. In this way, we obtain the point-cloud structured latent code $\mathbf{z} = [\mathbf{z}_x \oplus \mathbf{z}_h] \in \mathbb{R}^{(3+C_h) \times N}$ for flow matching.

**High-quality 3D Gaussian Decoding.** Given the point cloud-structured latent codes, how to decode them into high-quality 3D representation for supervision remains challenging. Though dense point cloud (Huang et al., 2024b) is a straightforward solution, it fails to depict high-quality 3D structure with limited point quantity. Here, we resort to surfel Gaussian (Huang et al., 2024a), an augmented point-based 3D representation that supports high-fidelity 3D surface modeling and efficient rendering. Specifically, our decoder first decodes the input through the 3D-DiT blocks (Peebles & Xie, 2023; Lan et al., 2024a), which has shown superior performance against traditional transformer layer:

$$\tilde{\mathbf{z}} := \mathcal{D}_T(\text{MLP}(\mathbf{z})), \tag{3}$$

where an MLP layer first projects the input latent to the corresponding dimension, and $\mathcal{D}_T$ is the DiT transformer. Since dense Gaussians are preferred for high-quality splatting (Kerbl et al., 2023), we gradually upsample the latent features through transformer blocks. Specifically, given a learnable embedding $\mathbf{z}_u \in \mathbb{R}^{f_u \times C}$ where $f_u$ is the up-sampling ratio, we prepend it to each token in the latent sequence. Then, $H$ layers of transformer blocks are used to model the upsampling process:

$$\mathbf{z}_i^{(k+1)} := \mathcal{D}_U^k([\mathbf{z}_u \oplus \tilde{\mathbf{z}}_i]), \tag{4}$$

where $\mathcal{D}_U^k$ is a transformer block for predicting the $k-$th levels of details (LoD) Gaussian as shown in Fig. 1, and $\mathbf{z}_i^{(k+1)} \in \mathbb{R}^{f_u \times C}$ are the upsampled set of tokens. The overall tokens $\mathbf{z}^{(k+1)} \in \mathbb{R}^{(f_u \times N) \times C}$ after up-sampling are used to predict the 13-dim attributes of surfel Gaussians.

To achieve denser Gaussians prediction, we cascade the upsampling transformer defined in Eq. (4) for $K$ times, giving the final Upsampler $\mathcal{D}_U$ for high-quality Gaussian output. Note that our solution outputs a set of Gaussians that are uniformly distributed on the 3D object surface with near $100\%$ Gaussian utilization ratio. Existing pixel-aligned Gaussian prediction models (Tang et al., 2024; Yinghao et al., 2024; Szymanowicz et al., 2023), however, usually waste $50\%$ Gaussians due to view overlaps and empty background color. Besides, our intermediate Gaussians output naturally serves as $K$ LoDs (Takikawa et al., 2021), which can be used in different scenarios to balance the rendering speed and quality.

**Training.** Our 3D VAE model is end-to-end optimized across both input views and randomly chosen views, minimizing image reconstruction objectives between the splatting renderings and ground-truth renderings. Besides image reconstruction loss, we also impose loss over geometry regularizations, KL constraints, and adversarial loss:

$$\mathcal{L}(\boldsymbol{\phi}, \boldsymbol{\psi}) = \mathcal{L}_{\text{render}} + \mathcal{L}_{\text{geo}} + \lambda_{\text{kl}}\mathcal{L}_{\text{KL}} + \lambda_{\text{GAN}}\mathcal{L}_{\text{GAN}}, \tag{5}$$

where $\mathcal{L}_{\text{render}}$ is a mixture of the $\mathcal{L}_1$ and VGG loss (Zhang et al., 2018), $\mathcal{L}_{\text{geo}}$ improves geometry reconstruction (Huang et al., 2024a), $\mathcal{L}_{\text{KL}}$ is the *KL-reg* loss (Kingma & Welling, 2013; Rombach et al., 2022) to regularize a structured latent space, and $\mathcal{L}_{\text{GAN}}$ improves perceptual fidelity. All loss terms except $\mathcal{L}_{\text{KL}}$ are applied over a randomly chosen LoD in each iteration, and the $\mathcal{L}_{\text{render}}$ is applied to both input-view and randomly sampled novel-view images. For details of geometry loss $\mathcal{L}_{\text{geo}}$, please refer to Sec. A.1.

## 3.2 CASCADED 3D GENERATION WITH FLOW MATCHING

After training the point-cloud structured 3D VAE, we get a dataset of $D$ shapes paired with condition vectors (*e.g.*, caption or images), $\{(\mathbf{z}_i, c_i)\}_{i \in [D]}$, where the shape is represented by latent code $\mathbf{z}$

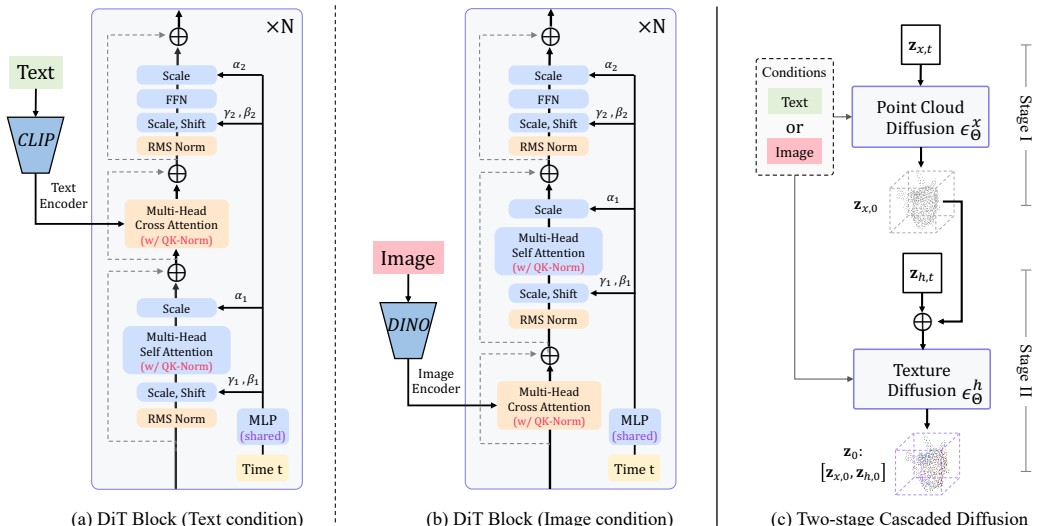

Figure 2: **Diffusion training of GAUSSIANANYTHING.** Based on the point-cloud structure 3D VAE, we perform cascaded 3D diffusion learning given text (a) and image (b) conditions. We adopt DiT architecture with AdaLN-single (Chen et al., 2023) and QK-Norm (Dehghani et al., 2023; Esser et al., 2021). For both condition modality, we send in the conditional feature with cross attention block, but at different positions. The 3D generation is achieved in two stages (c), where a point cloud diffusion model first generates the 3D layout $\mathbf{z}_{x,0}$, and a texture diffusion model further generates the corresponding point-cloud features $\mathbf{z}_{h,0}$. The generated latent code $\mathbf{z}_0$ is decoded into the final 3D object with the pre-trained VAE decoder.

through the 3D VAE aforementioned. Our goal is to train a flow-matching generative model to learn a flow-based prior on top of it. Below we present how we adapt flow-based models to our case.

**Cascaded Flow Matching over Symmetric Data.** As detailed in Sec. A.3, flow matching involves training a neural network $\boldsymbol{\epsilon}_\Theta$ to predict the velocity $v$ of the noisy input $\mathbf{z}_t$ with the straight-line trajectory. After training, $\boldsymbol{\epsilon}_\Theta$ can sample from a standard Normal prior $\mathcal{N}(0, I)$ by solving the reverse ODE/SDE (Karras et al., 2022). In our case, the training data point is the point-cloud structured latent code $\mathbf{z} = [\mathbf{z}_x \oplus \mathbf{z}_h] \in \mathbb{R}^{(3+C_h)\times N}$, which is symmetric and *permutation invariant* (Zeng et al., 2022; Nichol et al., 2022). Based on this property, we opt for diffusion transformer (Peebles & Xie, 2023) without positional encoding as the $\boldsymbol{\epsilon}_\Theta$ parameterization.

Here, rather than modeling $\mathbf{z}_x$ and $\mathbf{z}_h$ jointly, we empirically found that a cascaded framework (Ho et al., 2021; Lyu et al., 2024; 2023; Schröppel et al., 2024) leads to better performance. Specifically, a conditioned sparse point cloud generative model $\boldsymbol{\epsilon}_\Theta^x$ is first trained to generate the overall structure of the given object:

$$\mathcal{L}_w^x(x_0) = -\frac{1}{2}\mathbb{E}_{t\sim\mathcal{U}(t),\epsilon\sim\mathcal{N}(0,I)}\left[w_t^{\text{FM}}\lambda_t'\|\boldsymbol{\epsilon}_\Theta^x(\mathbf{z}_{x,t},t,c)-\epsilon\|^2\right] , \qquad (6)$$

and a point cloud feature generative model $\boldsymbol{\epsilon}_\Theta^h$ is cascaded to learn the corresponding $KL$-regularized feature conditioned on the sparse point cloud:

$$\mathcal{L}_w^h(x_0) = -\frac{1}{2}\mathbb{E}_{t\sim\mathcal{U}(t),\epsilon\sim\mathcal{N}(0,I)}\left[w_t^{\text{FM}}\lambda_t'\|\boldsymbol{\epsilon}_\Theta^h(\mathbf{z}_{h,t},\mathbf{z}_x,t,c)-\epsilon\|^2\right] . \qquad (7)$$

The detailed cascading process is detailed Fig. 2 (c). Our proposed design enables better geometry-texture disentanglement and facilitates 3D editing over specific shape properties. For derivations of the flow matching training objective, please refer to Sec. A.3 for more details.

**Conditioning Mechanism.** Compared to LRM (Hong et al., 2024b; Tang et al., 2024) line of work which intrinsically relies on image(s) as the input, our native flow-based method enables more flexible 3D generation from diverse conditions. As shown in Fig. 2 (a-b), for the text-conditioned model, we adopt CLIP (Radford et al., 2021) to extract *penultimate* tokens as the condition embeddings; and

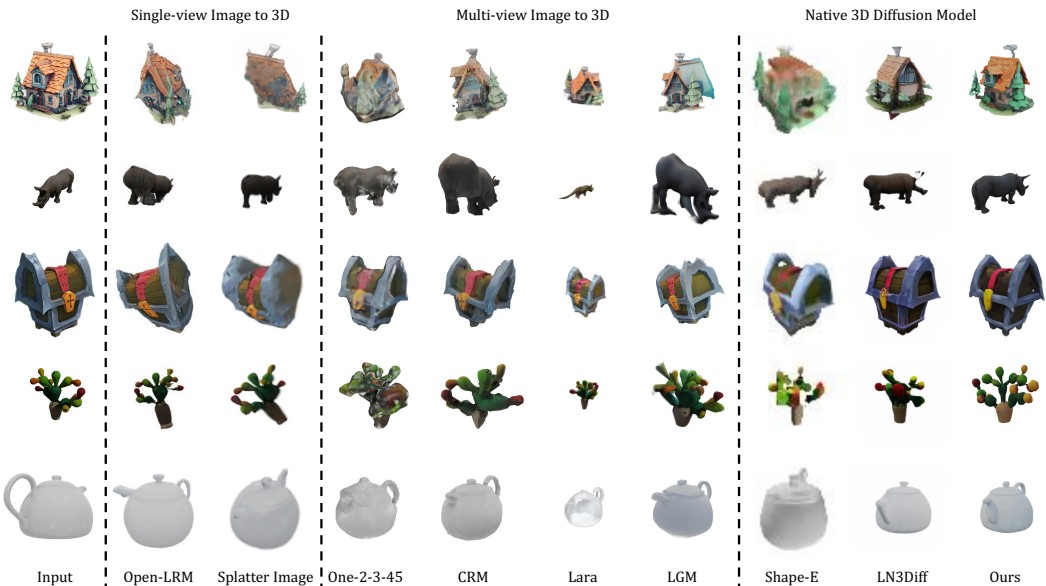

Figure 3: **Qualitative Comparison of Image-to-3D**. We showcase the novel view 3D reconstruction of all methods given a single image from unseen GSO dataset. Our proposed method achieves consistently stable performance across all cases. Note that though feed-forward 3D reconstruction methods achieve sharper texture reconstruction, these method fail to yield intact 3D predictions under challenging cases (e.g., the rhino in row 2). In contrast, our proposed native 3D diffusion model achieve consistently better performance. Better zoom in.

for the image conditioned model, we use DINOv2 (Oquab et al., 2023) to extract global and patch features. All conditions are injected into the DiT architecture through a pre-norm (Xiong et al., 2020) cross-attention block following the common practice of Zhang et al. (2024b). All the models are trained with Classifier-free Guidance (CFG) (Ho, 2021) by randomly dropping the conditions with a probability of $10\%$.

To cascade two flow-based models, we encode the output of stage-1 model $\epsilon_\Theta^x$ with $\mathrm{PE}(\mathbf{z}_x)$ as in Eq. (2), and add it to the first-layer features of $\epsilon_\Theta^h$. This guarantees that generated features are paired with the input sparse point cloud structure.

## 4 EXPERIMENTS

**Datasets.** To train our 3D VAE, we use the renderings provided by G-Objaverse (Qiu et al., 2023; Deitke et al., 2023b) and choose a high-quality subset with around $176K$ 3D instances, where each consists of $40$ random views with RGB, normal, depth map and camera pose. For text-conditioned diffusion training, we use the caption provided by Cap3D (Luo et al., 2023; 2024) and 3DTopia Hong et al. (2024a). For image-conditioned training, we randomly select an image in the dataset of the corresponding 3D instance as the condition.

**Implementation Details.** For 3D VAE, we choose $V = 8$ views of RGB-D-N renderings as input to guarantee a thorough coverage of the 3D object. The CNN Encoder is implemented with a similar architecture as LDM VAE (Rombach et al., 2022) with a down-sampling factor of $f = 8$, and the multi-view transformer has five layers as in RUST (Sajjadi et al., 2023). The sparse point cloud $\mathbf{z}_x$ has a size of $N \times 3$ where $N = 768$, and the corresponding featured point cloud $\mathbf{z}_h$ has a dimension of $N \times 10$. For upsampling blocks, we employ $K = 3$ blocks with $f_u^1 = 8$, $f_u^1 = 4$, and $f_u^1 = 3$, giving $73,768$ Gaussians in total. All transformer blocks follow a pre-norm design (Xiong et al., 2020). During 3D VAE training, the model is supervised by randomly chosen LoD renderings, with $\lambda_{\mathrm{kl}} = 2e - 6$, $\lambda_{\mathrm{d}} = 1000$, $\lambda_{\mathrm{n}} = 0.2$, and $\lambda_{\mathrm{GAN}} = 0.1$. We adopt batch size $64$ with both input and random novel views for training. During the conditional flow-based model training stage, we adopt batch size $256$. All models are efficiently and stably trained with lr $= 1e - 4$ on $8 \times$A100 GPUs for $1M$ iterations with BF16 and FlashAttention (Dao, 2024) enabled. We use CFG=4 and $250$ ODE steps for all sampling results.

Table 2: **Quantitative evaluation of image-conditioned 3D generation.** Here, quality of both 2D rendering and 3D shapes is evaluated. As shown below, the proposed method demonstrates strong performance across all metrics. Although multi-view images-to-3D approaches like LGM achieves better performance on the FID/KID metrics, they fall short on more advanced image quality assessment metrics such as CLIP-I and MUSIQ. Besides, they perform significantly worse in 3D shape quality. For multi-view to 3D baselines, we also include the number of input views (V=#).

| Method | CLIP-I↑ | FID↓ | KID(%)↓ | MUSIQ↑ | P-FID↓ | P-KID(%)↓ | COV(%)↑ | MMD(‰)↓ |
|---|---|---|---|---|---|---|---|---|
| OpenLRM | 86.37 | 38.41 | 1.87 | 45.46 | 35.74 | 12.60 | 39.33 | 29.08 |
| Splatter-Image | 84.10 | 48.80 | 3.65 | 30.33 | 19.72 | 7.03 | 37.66 | 30.69 |
| One-2-3-45 (V=12) | 80.72 | 88.39 | 6.34 | 59.02 | 72.40 | 30.83 | 33.33 | 35.09 |
| CRM (V=6) | 85.76 | 45.53 | 1.93 | 64.10 | 35.21 | 13.19 | 38.83 | 28.91 |
| Lara (V=4) | 84.64 | 43.74 | 1.95 | 39.37 | 32.37 | 12.44 | 39.33 | 28.84 |
| LGM (V=4) | 87.99 | 19.93 | 0.55 | 54.78 | 40.17 | 19.45 | 50.83 | 22.06 |
| Shape-E | 77.05 | 138.53 | 11.95 | 31.51 | 20.98 | 7.41 | 61.33 | 19.17 |
| LN3Diff | 87.24 | 29.08 | 0.89 | 50.39 | 27.17 | 10.02 | 55.17 | 19.94 |
| **Ours** | 89.06 | 24.21 | 0.76 | 65.17 | 8.72 | 3.22 | 59.50 | 15.48 |

## 4.1 METRICS AND BASELINES

**Evaluating Image-to-3D Generation.** We evaluate GAUSSIANANYTHING on both image and text conditioned generation. Regarding image-conditioned 3D generation methods, we compare the proposed method with three lines of methods: *single-image to 3D methods*: OpenLRM (He & Wang, 2023; Hong et al., 2024b), Splatter Image (Szymanowicz et al., 2023), *multi-view images to 3D methods*: One-2-3-45 Liu et al. (2023a), CRM (Wang et al., 2024), Lara (Chen et al., 2024a), LGM (Tang et al., 2024), and *native 3D diffusion models*: LN3Diff-image (Lan et al., 2024a).

Quantitatively, we benchmark rendering metrics with CLIP-I Radford et al. (2021), FID (Heusel et al., 2017), KID (Bińkowski et al., 2018), and MUSIQ-koniq (Ke et al., 2021; Zhou et al., 2022). For 3D quality metrics, we benchmark Point cloud FID (P-FID), Point cloud KID (P-KID), Coverage Score (COV), and Minimum Matching Distance (MMD). Following previous works Nichol et al. (2022); Zhang et al. (2023); Yariv et al. (2024), we adopt the pre-trained PointNet++ provided by Point-E (Nichol et al., 2022) for calculating P-FID and K-FID. Qualitatively, GSO (Downs et al., 2022; Zheng & Vedaldi, 2023) dataset is used for visually inspecting image-conditioned generation.

**Evaluating Text-to-3D Generation.** Regarding text-conditioned 3D generation methods, we compare against Point-E (Nichol et al., 2022), Shape-E (Jun & Nichol, 2023), 3DTopia (Hong et al., 2024a), and LN3Diff-text (Lan et al., 2024a). CLIP score (Radford et al., 2021) is reported following the previous works (Lan et al., 2024a; Hong et al., 2024a), with aesthetic scores MUSIQ-AVA (Ke et al., 2021) and Q-Align (Wu et al., 2023b) also included.

Table 1: **Quantitative Evaluation on Text-to-3D.** The proposed method outperforms competitive alternatives on both CLIP scores and aesthetic scores.

| Method | ViT-B/32↑ | ViT-L/14↑ | MUSIQ-AVA ↑ | Q-Align ↑ |
|---|---|---|---|---|
| Point-E | 26.35 | 21.40 | 4.08 | 1.21 |
| Shape-E | 27.84 | 25.84 | 3.69 | 1.56 |
| LN3Diff | 29.12 | 27.80 | 4.16 | 2.22 |
| 3DTopia | 30.10 | 28.11 | 3.31 | 1.42 |
| Ours | **31.80** | **29.38** | **4.99** | **3.13** |

## 4.2 EVALUATION

In this section, we evaluate our proposed method over image-to-3D generation, text-to-3D generation, and 3D-aware editing. Please check the appendix for more visual results and point cloud-conditioned 3D generation. **Image-to-3D Generation.** Our proposed framework enables 3D generation given single-view image conditions, leveraging the architecture detailed in Fig. 2 (b). Following current method (Chen et al., 2024a; Tang et al., 2024), we qualitatively benchmark our method in Fig. 3 over the single-view 3D reconstruction task on the unseen images from the GSO dataset. Our proposed framework is robust to inputs with complicated structures (row 1,3,4) and self-occlusion (row 2,5), yielding consistently intact 3D reconstruction. Besides, our generative-based method shows a more natural back-view reconstruction, as opposed to regression-based methods that are commonly blurry on uncertain areas.

Quantitatively, we showcase the evaluation in Tab. 2. As can be seen, our proposed method achieves state-of-the-art performance over CLIP-I and all 3D metrics, with competitive results over conventional 2D rendering metrics FID/KID. Note that LGM leverages pre-trained MVDream (Shi et al., 2023b) as the first-stage generation, and then maps the generated 4 views to pixel-aligned 3D Gaussians. This cascaded pipeline achieves better visual quality, but prone to yield distorted 3D geometry, as visualized in Fig. 3.

**Text-to-3D Generation.** We demonstrate the text-to-3D generation performance in Fig. 4 and Tab. 1. The flow-based model trained on GAUSSIANANYTHING's latent space has demonstrated high-quality text-to-3D generation of generic 3D objects, yielding superior performance in terms of object structure, textures, and surface normals. Quantitatively, our proposed method achieves better text-3D alignment against competitive baselines.

**3D-aware Editing.** Compared to existing methods that use unstructured tokens for 3D diffusion learning (Jun & Nichol, 2023), our proposed point-cloud structured latent space naturally facilitates geometry-texture disentanglement and allows for interactive 3D editing. As visualized in Fig. 5, given the text-conditioned generated point cloud $\mathbf{z}_0$ by $\epsilon_\Theta^x$, we sample the final 3D objects with $\epsilon_\Theta^h$ with a different random seed. As can be seen, the generated 3D objects maintain a consistent structure layout while yielding diverse textures. Besides, by directly manipulating the conditioned point cloud $\mathbf{z}_{x,0}$, our proposed method enables interactive 3D editing, as in 2D models (Pan et al., 2023; Mou et al., 2023b). This functionality greatly facilitates the 3D content creation process for artists and opens up new possibilities for 3D editing with diffusion models.

### 4.3 ABLATION STUDY AND ANALYSIS

Table 3: **Ablation of 3D VAE Design.** We ablate the design of our 3D VAE. Input-side, leveraging multi-view RGB-D-N renderings shows superior performance against dense point cloud. Besides, adding Gaussian up-sampling modules leads to consistent performance gain.

| Design | LPIPS@100K |
|---|---|
| Dense PCD as Input | 0.174 |
| Multi-view RGB-D as Input | 0.163 |
| + Normal Map | 0.157 |
| + Gaussian SR Module | 0.095 |
| + 3 × Gaussian SR Module | **0.067** |

Table 4: **Gaussian Utilization Ratio.** We compare the effective Gaussians (opacity $> 0.005$) used during splatting here. Pixel-aligned Gaussian prediction methods waste a large portion of Gaussians when representing 3D object due to white background and multi-view overlap, while our proposed Gaussian predictions yields more compact reconstruction results.

| Method | High-opacity Gaussians (%) |
|---|---|
| Splatter Image | 17.14 |
| LGM | 52.63 |
| Ours | **96.84** |

**3D VAE Design.** In Tab. 3, we benchmark each component of our 3D VAE architecture over a subset of Obiaverse with $50K$ instances and record the LPIPS at $100K$ iterations. As shown in Tab. 3, our

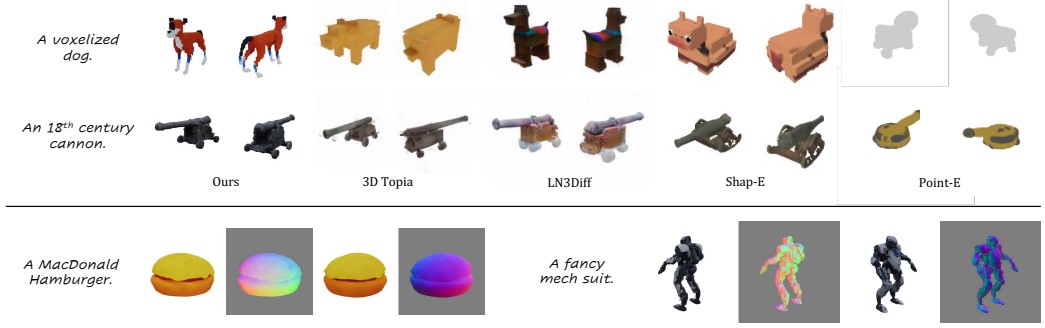

Figure 4: **Qualitative Comparison of Text-to-3D**. We present text-conditioned 3D objects generated by GAUSSIANANYTHING, displaying two views of each sample. The top section compares our results with baseline methods, while the bottom shows additional samples from our method along with their geometry maps. Our approach consistently yields better quality in terms of geometry, texture, and text-3D alignment.

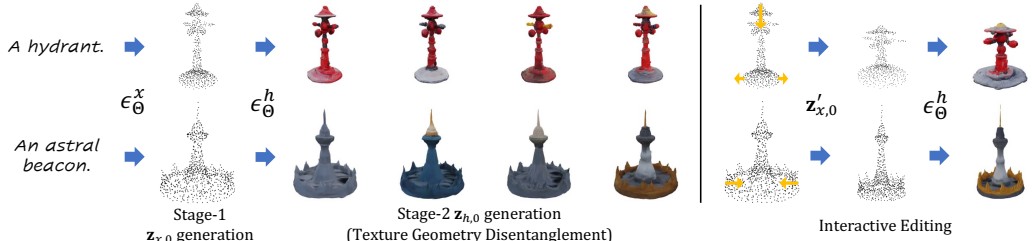

Figure 5: **3D editing.** Given two text prompts, we generate the corresponding point cloud $\mathbf{z}_{0,x}$ with stage-1 diffusion model with $\boldsymbol{\epsilon}_\Theta^x$, and the corresponding point cloud features $\mathbf{z}_{0,h}$ can be further generated with $\boldsymbol{\epsilon}_\Theta^h$. As can be seen, the samples from stage-2 are consistent in overall 3D structures but with diverse textures. Thanks to the proposed Point Cloud-structured Latent space, our method supports interactive 3D structure editing. This is achieved by first modifying the stage-1 point cloud $\mathbf{z}_{0,x} \rightarrow \mathbf{z}'_{0,x}$, and then regenerate the 3D object with the same Gaussian noise.

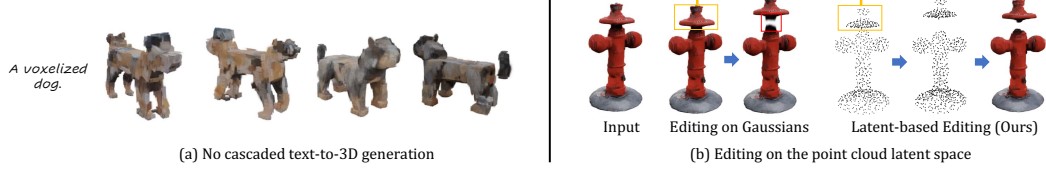

(a) No cascaded text-to-3D generation    (b) Editing on the point cloud latent space

Figure 6: **Qualitative ablation of Cascaded diffusion and latent space editing.** We first show the effectiveness of our two-stage cascaded diffusion framework in (a). Compared to Fig. 4, the single-stage 3D diffusion yields worse texture details and 3D structure intactness. In (b), we disjoint the hydrant cover to demonstrate that our latent point cloud editing yields smoother and more reasonable results, while direct editing on 3D Gaussians shows tearing artifacts.

input design performs better against dense $(16, 384)$ colored point cloud (Zhang et al., 2024b), and the reconstruction quality consistently improves by including normal map as input and cascading more Gaussian upsampling blocks.

**Gaussian Utilization Ratio.** Besides, we showcase a high Gaussian utilization ratio of our proposed method. Specifically, we calculate the ratio of Gaussians with an opacity greater than $0.005$ as *effective* Gaussians, as they contribute well to the final rendering. We calculate the statistics over $50K$ 3D instances. As shown in Tab. 4, our proposed Gaussian prediction framework achieves a much higher utilization ratio. On the contrary, pixel-aligned Gaussian prediction models waste a noticeable portion of Gaussians on the overlapping views and white backgrounds.

**Effectiveness of Cascaded 3D Diffusion.** We qualitatively ablate the cascaded model design in Fig. 6 (a), where a single text-conditioned DiT is trained to synthesize the 3D point cloud and features jointly. Clearly, the jointly trained model has a worse texture with 3D shape artifacts to our cascaded design. Besides bringing better editing capability as shown in Fig. 5, our cascaded design enables more flexible training, where the models of two stages can be trained in parallel.

**3D Editing on the 3D Latent Space.** Finally, we ablate the 3D editing performance in Fig. 6 (b). As can be seen, direct editing on the final Gaussians leads to 3D artifacts, while editing on our 3D latent space yields more holistic and cleaner results since suitable features are re-generated after editing. Besides, our method enables easy editing on the sparse point cloud, compared to directly manipulating dense 3D Gaussians (Dong et al., 2024).

## 5 CONCLUSION AND DISCUSSIONS

In this work, we present a new paradigm of 3D generative model by learning the diffusion model over a interactive 3D latent space. A dedicated 3D variational autoencoder encodes multi-view 3D attributes renderings into a point-cloud structured latent space, where multi-modal diffusion learning can be efficiently performed. Our framework achieves superior performance over both text- and image-conditioned 3D generation, and potentially facilitates numerous downstream applications in 3D vision and graphics tasks. Please check the appendix for the discussion of limitations.

**Acknowledgement.** This study is supported under the RIE2020 Industry Alignment Fund Industry Collaboration Projects (IAF-ICP) Funding Initiative, as well as cash and in-kind contributions from the industry partner(s). It is also supported by Singapore MOE AcRF Tier 2 (MOE-T2EP20221-0011) and National Research Foundation, Singapore, under its NRF Fellowship Award NRF-NRFF16-2024-0003.

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

APPENDIX

# A IMPLEMENTATION DETAILS

## A.1 TRAINING DETAILS

**VAE Architecture.** For the convolutional encoder $\mathcal{E}_\phi$, we adopt a lighter version of LDM Rombach et al. (2022) encoder with channel 64 and 1 residual blocks for efficiency. When training on Objaverse with $V = 8$, we incorporate 3D-aware attention Shi et al. (2023b) in the middle layer of the convolutional encoder. The multi-view transformer architecture is similar to RUST (Sajjadi et al., 2023; 2022). For each upsampler $\mathcal{D}_U^k$, we have 2 transformer blocks in the middle. All hyperparameters remain at their default settings. Regarding the transformer decoder $\mathcal{D}_T$, we employ the DiT-B/2 architecture due to VRAM constraints. Compared to LN3Diff (Lan et al., 2024a), we do not adopt cross-plane attention in the transformer decoder.

**Diffusion Model.** We mainly adopt the diffusion training pipeline implementation from SiT Ma et al. (2024), with pred-$v$ objective, GVP schedule, and uniform $t$ sampling. ODE solver with 250 steps is used for all the results shown in the paper. For the DiT architecture with cross attention and single-adaLN-zero design, we mainly refer to PixArt Chen et al. (2023). The diffusion transformer is built with 24 layers with 16 heads and 1024 hidden dimension, which result in 458M parameters. For all the diffusion models, we further add the global token to $t$ features as part of the condition input.

**Details of Geometry Loss in VAE Training.** The geometry loss $\mathcal{L}_{\text{geo}}$ is composed of two regularization terms, including the depth distortion loss to concentrate the weight distribution along rays, inspired by Mip-NeRF (Barron et al., 2021; 2022). Given a ray of pixel, the distortion loss is defined as

$$\mathcal{L}_d = \sum_{i,j} \omega_i \omega_j |d_i - d_j|, \tag{8}$$

where $\omega_i = \alpha_i \hat{\mathcal{G}}_i(\mathbf{u}(\mathbf{x})) \prod_{j=1}^{i-1}(1 - \alpha_j \hat{\mathcal{G}}_j(\mathbf{u}(\mathbf{x})))$ is the blending weight of the $i-$th intersection and $d_i$ is the depth of the intersection points. Besides, as surfel Gaussians explicitly model the primitive normals, we encourage the splats' normal to locally approximate the actual object surface:

$$\mathcal{L}_n = \sum_i \omega_i (1 - \hat{N}_i^{\text{T}} N), \tag{9}$$

where $\hat{N}$ is the predicted normal maps. The final geometry loss is given by $\mathcal{L}_{\text{geo}} = \lambda_{\text{d}} \mathcal{L}_d + \lambda_{\text{n}} \mathcal{L}_n$.

## A.2 DATA AND BASELINE COMPARISON

**Training Data.** For Objaverse, we use a high-quality subset from the pre-processed rendering from G-buffer Objaverse Qiu et al. (2023) for experiments. Since G-buffer Objaverse splits the subset into 10 general categories, we use all the 3D instances except from "Poor-quality": Human-Shape, Animals, Daily-Used, Furniture, Buildings&Outdoor, Transportations, Plants, Food and Electronics. The ground truth camera pose, rendered multi-view images, normal, depth maps, and camera poses are used for stage-1 VAE training.

**Details about Baselines.** We use the official released code and checkpoint for all the comparisons shown in the paper. For the evaluation on the GSO dataset, we use the rendering provided by Free3D (Zheng & Vedaldi, 2023).

**Evaluation details.** For quantitative benchmark in Tab. 2, we use 600 instances from Objaverse with ground truth 3D mesh for evaluation. To calculate the visual metrics (FID/KID/MUSIQ), we use the first rendered instance as the image condition and render 24 images with fixed elevation (+15 degrees) with uniform azimuths trajectory ($24 \times 15$ degrees) with radius= 1.8. For 3D metrics, we export the extracted 3D mesh and sample 4096 points using FPS sampling on the mesh surface. The ground truth surface point cloud is processed in the same way. The pre-trained PointNet++ model from Point-E is used for P-FID and P-KID evaluation. All generated 3D models are aligned into the same canonical space before 3D metrics calculation. All intermediate results of the baselines for evaluation will be released.

### A.3 MORE PRELIMINARIES

**2D Gaussian Splatting (2DGS).** Since 3DGS (Kerbl et al., 2023) models the entire angular radiance in a blob, it fails to reconstruct high-quality object surfaces. To resolve this issue, Huang et al. (2024a) proposed 2DGS (surfel-based GS) that simplifies the 3-dimensional modeling by adopting "flat" 2D Gaussians embedded in 3D space, which enables better alignment with thin surfaces.

Notation-wise, the 2D splat is characterized by its central point $\mathbf{p}_k$, two principal tangential vectors $\mathbf{t}_u$ and $\mathbf{t}_v$, and a scaling vector $\mathbf{S} = (s_u, s_v)$ that controls the variances of the 2D Gaussian. Notice that the primitive normal is defined by two orthogonal tangential vectors $\mathbf{t}_w = \mathbf{t}_u \times \mathbf{t}_v$. Thus, the 2D Gaussian is parameterized with

$$P(u, v) = \mathbf{p}_k + s_u \mathbf{t}_u u + s_v \mathbf{t}_v v = \mathbf{H}(u, v, 1, 1)^{\mathrm{T}} \tag{10}$$

$$\text{where } \mathbf{H} = \begin{bmatrix} s_u \mathbf{t}_u & s_v \mathbf{t}_v & \mathbf{0} & \mathbf{p}_k \\ 0 & 0 & 0 & 1 \end{bmatrix} = \begin{bmatrix} \mathbf{RS} & \mathbf{p}_k \\ \mathbf{0} & 1 \end{bmatrix} \tag{11}$$

Where $\mathbf{H}$ parameterizes the local 2D Gaussian geometry. For the point $\mathbf{u} = (u, v)$ in $uv$ space, its 2D Gaussian value can then be evaluated by standard Gaussian $\mathcal{G}(\mathbf{u}) = \exp\left(-\frac{u^2 + v^2}{2}\right)$, and the center $\mathbf{p}_k$, scaling $(s_u, s_v)$, and the rotation $(\mathbf{t}_u, \mathbf{t}_v)$ are all learnable parameters. Following 3DGS Kerbl et al. (2023), each 2D Gaussian primitive has opacity $\alpha$ and view-dependent appearance $\mathbf{c}$, and can be rasterized via volumetric alpha blending:

$$\mathbf{c}(\mathbf{x}) = \sum_{i=1} \mathbf{c}_i \, \alpha_i \, \hat{\mathcal{G}}_i(\mathbf{u}(\mathbf{x})) \prod_{j=1}^{i-1} (1 - \alpha_j \, \hat{\mathcal{G}}_j(\mathbf{u}(\mathbf{x}))), \tag{12}$$

where the integration process is terminated when the accumulated opacity reaches saturation. During optimization, pruning and densification operations are iteratively applied.

**Flow Matching and Diffusion Model.** Diffusion models create data from noise (Song et al., 2021) and are trained to invert forward paths of data towards random noise. The forward path is constructed as $z_t = a_t x_0 + b_t \epsilon$, where $\epsilon \sim \mathcal{N}(0, I)$, $a_t$ and $b_t$ are hyper parameters. The choice of forward process has proven to have important implications for the backward process of data sampling (Lin et al., 2023).

Recently, flow matching (Liu et al., 2023c; Albergo et al., 2023; Lipman et al., 2023) has introduced a particular choice for the forward path, which has better theoretical properties and has been verified on the large-scale study (Esser et al., 2024). Given a unified diffusion objective (Karras et al., 2022):

$$\mathcal{L}_w(x_0) = -\frac{1}{2} \mathbb{E}_{t \sim \mathcal{U}(t), \epsilon \sim \mathcal{N}(0,I)} \left[ w_t \lambda_t' \| \epsilon_\Theta(z_t, t) - \epsilon \|^2 \right], \tag{13}$$

where $\lambda_t := \log \frac{a_t^2}{b_t^2}$ denotes *signal-to-noise ratio*, and $\lambda_t'$ denotes its derivative. By setting $w_t = \frac{t}{1-t}$ with $z_t = (1 - t)x_0 + t\epsilon$, flow matching defines the forward process as a straight path between the data distribution and the Normal distribution. The network $\epsilon_\Theta$ directly predicts the *velocity* $v_\Theta$, and please check the following section for more detailed derivation.

**Derivation of the Training Objective of Flow Matching.** Since three works (Liu et al., 2023c; Albergo et al., 2023; Lipman et al., 2023) proposed the flow matching idea simultaneously, we adopt the unified formulation defined in Esser et al. (2024) in Eq. 6 and Eq. 7. Here we brief the background of conditional flow matching, and please read the Sec.2 of Esser et al. (2024) for in-depth analysis.

Specifically, consider the forward diffusion process (Ho et al., 2020)

$$z_t = a_t x_0 + b_t \epsilon, \quad \text{where } \epsilon \sim \mathcal{N}(0, I). \tag{14}$$

To express the relationship between $z_t$, $x_0$, and $\epsilon$, we define the mappings $\psi_t$ and $u_t$ as:

$$\psi_t(\cdot \mid \epsilon) : x_0 \mapsto a_t x_0 + b_t \epsilon, \tag{15}$$

$$u_t(z \mid \epsilon) := \psi_t'\big(\psi_t^{-1}(z \mid \epsilon) \mid \epsilon\big), \tag{16}$$

where $\psi_t^{-1}$ and $\psi_t'$ are the inverse and derivative of $\psi_t$, respectively.

Since $z_t$ can be viewed as a solution to the ODE

$$z'_t = u_t(z_t \mid \epsilon), \quad \text{with initial condition } z_0 = x_0, \tag{17}$$

the conditional vector field $u_t(\cdot \mid \epsilon)$ generates the conditional probability path $p_t(\cdot \mid \epsilon)$.

Remarkably, one can construct a marginal vector field $u_t$ that generates the marginal probability paths $p_t$ (Lipman et al., 2023), using the conditional vector fields $u_t(\cdot \mid \epsilon)$:

$$u_t(z) = \mathbb{E}_{\epsilon \sim \mathcal{N}(0,I)} \left[ u_t(z \mid \epsilon) \frac{p_t(z \mid \epsilon)}{p_t(z)} \right]. \tag{18}$$

The marginal vector field $u_t$ can be learned by minimizing the *Conditional Flow Matching* objective:

$$\mathcal{L}_{\text{CFM}} = \mathbb{E}_{t, p_t(z \mid \epsilon), p(\epsilon)} \big\| v_\Theta(z, t) - u_t(z \mid \epsilon) \big\|_2^2. \tag{19}$$

To make this objective explicit, we substitute:

$$\psi'_t(x_0 \mid \epsilon) = a'_t x_0 + b'_t \epsilon, \tag{20}$$

$$\psi_t^{-1}(z \mid \epsilon) = \frac{z - b_t \epsilon}{a_t}, \tag{21}$$

into the expression for $u_t(z \mid \epsilon)$:

$$z'_t = u_t(z_t \mid \epsilon) = \frac{a'_t}{a_t} z_t - \epsilon b_t \left( \frac{a'_t}{a_t} - \frac{b'_t}{b_t} \right). \tag{22}$$

Next, consider the *signal-to-noise ratio* $\lambda_t := \log \frac{a_t^2}{b_t^2}$. With $\lambda'_t = 2\big(\frac{a'_t}{a_t} - \frac{b'_t}{b_t}\big)$, the expression for $u_t(z_t \mid \epsilon)$ can be rewritten as:

$$u_t(z_t \mid \epsilon) = \frac{a'_t}{a_t} z_t - \frac{b_t}{2} \lambda'_t \epsilon. \tag{23}$$

Using this reparameterization, the $\mathcal{L}_{\text{CFM}}$ objective can be reformulated as a noise-prediction objective:

$$\mathcal{L}_{\text{CFM}} = \mathbb{E}_{t, p_t(z \mid \epsilon), p(\epsilon)} \left\| v_\Theta(z, t) - \frac{a'_t}{a_t} z + \frac{b_t}{2} \lambda'_t \epsilon \right\|_2^2 \tag{24}$$

$$= \mathbb{E}_{t, p_t(z \mid \epsilon), p(\epsilon)} \left( -\frac{b_t}{2} \lambda'_t \right)^2 \big\| \epsilon_\Theta(z, t) - \epsilon \big\|_2^2, \tag{25}$$

where we define:

$$\epsilon_\Theta := \frac{-2}{\lambda'_t b_t} \left( v_\Theta - \frac{a'_t}{a_t} z \right). \tag{26}$$

Since the optimal solution remains invariant to time-dependent weighting, one can derive various weighted loss functions that guide optimization towards the desired solution. For a unified analysis of different approaches, including classic diffusion formulations, we express the objective as Kingma & Gao (2023):

$$\mathcal{L}_w(x_0) = -\frac{1}{2} \mathbb{E}_{t \sim \mathcal{U}(t), \epsilon \sim \mathcal{N}(0,I)} \big[ w_t \lambda'_t \big\| \epsilon_\Theta(z_t, t) - \epsilon \big\|^2 \big], \tag{27}$$

where $w_t = -\frac{1}{2} \lambda'_t b_t^2$ corresponds to $w_t^{\text{FM}}$ used in Eq. 6 and Eq. 7.

# B  DISCUSSIONS OF LIMITATIONS

We acknowledge that the texture quality of our proposed method is still inferior to the state-of-the-art multi-view based 3D generative models, i.e., LGM. Besides, the visual quality of the text-conditioned 3D generation of native 3D generation methods is worse compared to SDS-based alternatives, despite being much faster and shows better diversity. We believe our method has made a step forward towards bridging the gap. To further improve the performance, we list some potential directions in the following:

1. **Enhancing the 3D VAE Quality..** The performance of the 3D VAE could be improved by increasing the number of latent points and incorporating a pixel-aligned 3D reconstruction paradigm, such as PiFU (Saito et al., 2019), to achieve finer-grained geometry and texture alignment.

2. **Incorporating Additional Losses in Diffusion Training.** Currently, the diffusion training relies solely on latent-space flow matching. Prior work, such as DMV3D (Xu et al., 2024c), demonstrates that incorporating a rendering loss can significantly enhance the synthesis of high-quality 3D textures. Adding reconstruction supervision during diffusion training is another promising avenue to improve output fidelity.

3. **Leveraging 2D Pre-training Priors.** At present, the models are trained exclusively on 3D datasets and do not utilize 2D pre-training priors as effectively as multi-view (MV)-based 3D generative models. A potential improvement is to incorporate 2D priors more effectively, for instance, by using multi-view synthesized images as conditioning during training instead of single-view images.

4. **Expanding Dataset Diversity.** Utilizing more diverse and extensive 3D datasets, such as Objaverse-XL (Deitke et al., 2023a) and MVImageNet (Yu et al., 2023), could further enhance the quality and generalizability of 3D generation.

5. **Support of Physics-based Rendering.** Currently, our proposed method leverages 2DGS as the underlying 3D representation since it balances the 2D rendering and 3D surface quality. However, incorporating more advanced 3D representations such as 3DGRT (Moenne-Loccoz et al., 2024) can further support physics-based ray tracing over 3D Gaussians. Besides, this can be achieved by directly fine-tuning our pre-trained 3D VAE decoder only with the PBR rendering pipeline.

By addressing these aspects, the proposed method could achieve significant advancements in both the quality and versatility of its 3D generation capabilities.

## C    DISCUSSIONS OF CONCURRENT WORK

After the submission of this paper, several related works on 3D generation have been released. We discuss them here.

FreeSplatter (Xu et al., 2024b) extends the multi-view generation and 3D reconstruction pipeline for 3D object generation, eliminating the need for input poses. It also supports 3D scene generation by training on the scene-level datasets (Yao et al., 2020; Dai et al., 2017). While following a different paradigm, our method could benefit from its pose-free design to enable image-conditioned 3D generation from multiple input views.

Geometric Distribution (Zhang et al., 2024a) introduces a novel 3D representation by learning a point cloud diffusion model for single-instance objects. This approach closely resembles our stage-1 diffusion model but focuses on encoding geometric information. While it shows strong potential for downstream tasks, its efficiency remains a challenge for broader applications.

AtlasGaussians (Yang et al., 2025), like our method, proposes a native 3D diffusion model over 3D Gaussians. It employs point clouds as input for 3D VAE compression and utilizes local patch features for 3D Gaussian decoding. However, it lacks an explicit 3D latent space, making it unsuitable for interactive 3D editing.

Trellis (Xiang et al., 2024) adopts a similar cascaded 3D flow matching framework and achieves remarkable results in object generation. Unlike our method, which employs a point cloud latent space, Trellis leverages a sparse voxel structure with efficient operators for high-quality and fast 3D generation. It also supports interactive 3D editing via its explicit latent space. We hope this design, along with our proposed method, can establish a canonical paradigm for future 3D native generative models.

## D    MORE VISUAL RESULTS AND VIDEOS

Please also check our supplementary *video demo* and *attached folders* for video results.

**Overview of the Qualitative Performance.** In Fig. 7, we include an overview of the qualitative

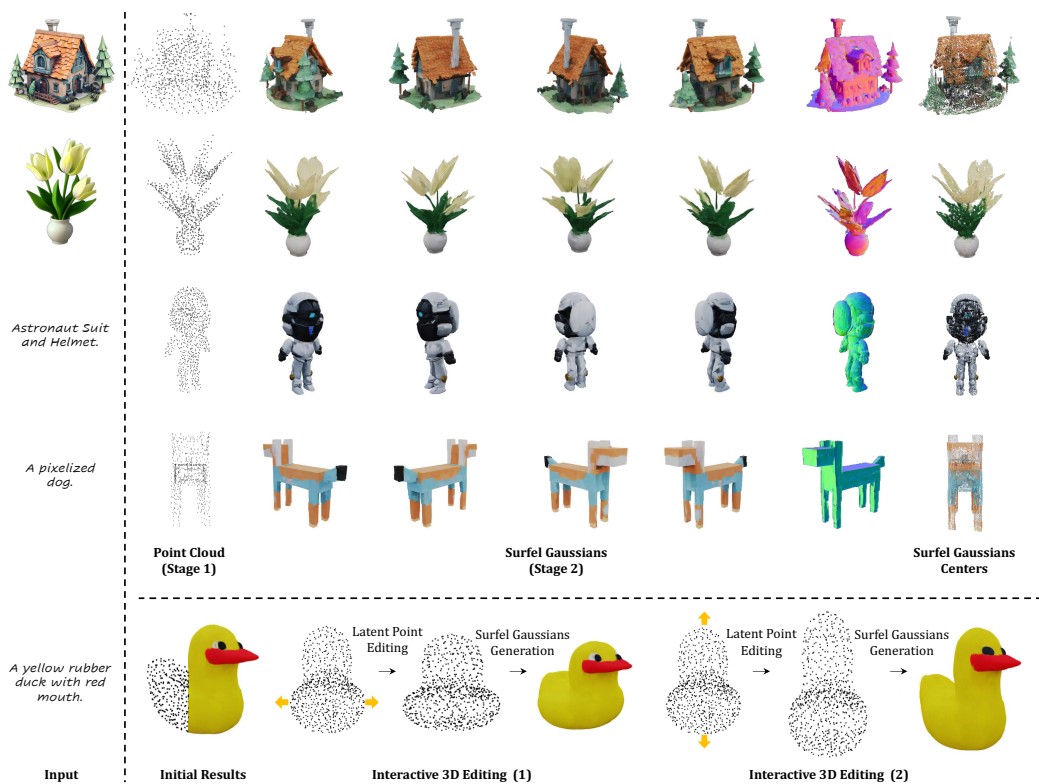

Figure 7: Our method generates *high-quality* and *editable* surfel Gaussians through a cascaded 3D diffusion pipeline, given single-view images or texts as the conditions.

performance of the proposed method, GAUSSIANANYTHING. Here, we show the single-view conditioned 3D generation, text-conditioned 3D generation, and 3D-aware editing capabilities.

**3D VAE Reconstruction.** In Fig. 8, we include the 3D VAE reconstruction results of our model at 3 level of details (LoD). Thanks to the versatile multi-view 3D attributes input and transformer design, our 3D VAE enables high-quality 3D reconstruction with visually attractive textures and smooth surface. The encoded point cloud-structured latent codes, $\mathbf{z}$, serves as a compact proxy for efficient 3D diffusion training. Besides, the 2D Gaussians Upsampler naturally facilitates LoD and facilitates speed / quality trade off in practice.

**More Text-to-3D results.** In Fig. 9, we present additional qualitative comparisons of text-to-3D generation with GAUSSIANANYTHING. For this evaluation, we use relatively complex captions as input conditions and display two random samples generated by our model. As shown, GAUSSIANANYTHING produces visually appealing results that characterized rich textures, smooth surface, and notable diversity. To further demonstrate the generality of our proposed method, in Fig. 10 we include the *uncurated* text-to-3D results over DF-415 (Poole et al., 2022) prompts with captions and more detailed descriptions.

**Point-to-3D Generation with Cascaded Point-E.** Moreover, the cascaded design of our stage-2 diffusion model, $\epsilon_\Theta^h$, enables flexible 3D generation given point clouds from diverse sources. To demonstrate this capability, we integrate the output of a state-of-the-art 3D point cloud generative model, such as Point-E (Nichol et al., 2022), into the GAUSSIANANYTHING generation pipeline. Specifically, we first generate the point cloud using Point-E based on a caption condition $c$. This generated point cloud is then used as input $\mathbf{z}_x$ to our stage-2 point cloud/text-conditioned diffusion model $\epsilon_\Theta^h$. As shown in Fig. 11, the generated surfel Gaussians exhibit significantly improved texture quality and geometry fidelity compared to the Point-E point cloud outputs. This capability broadens the applicability of our method, enabling it to benefit from recent advances in point cloud

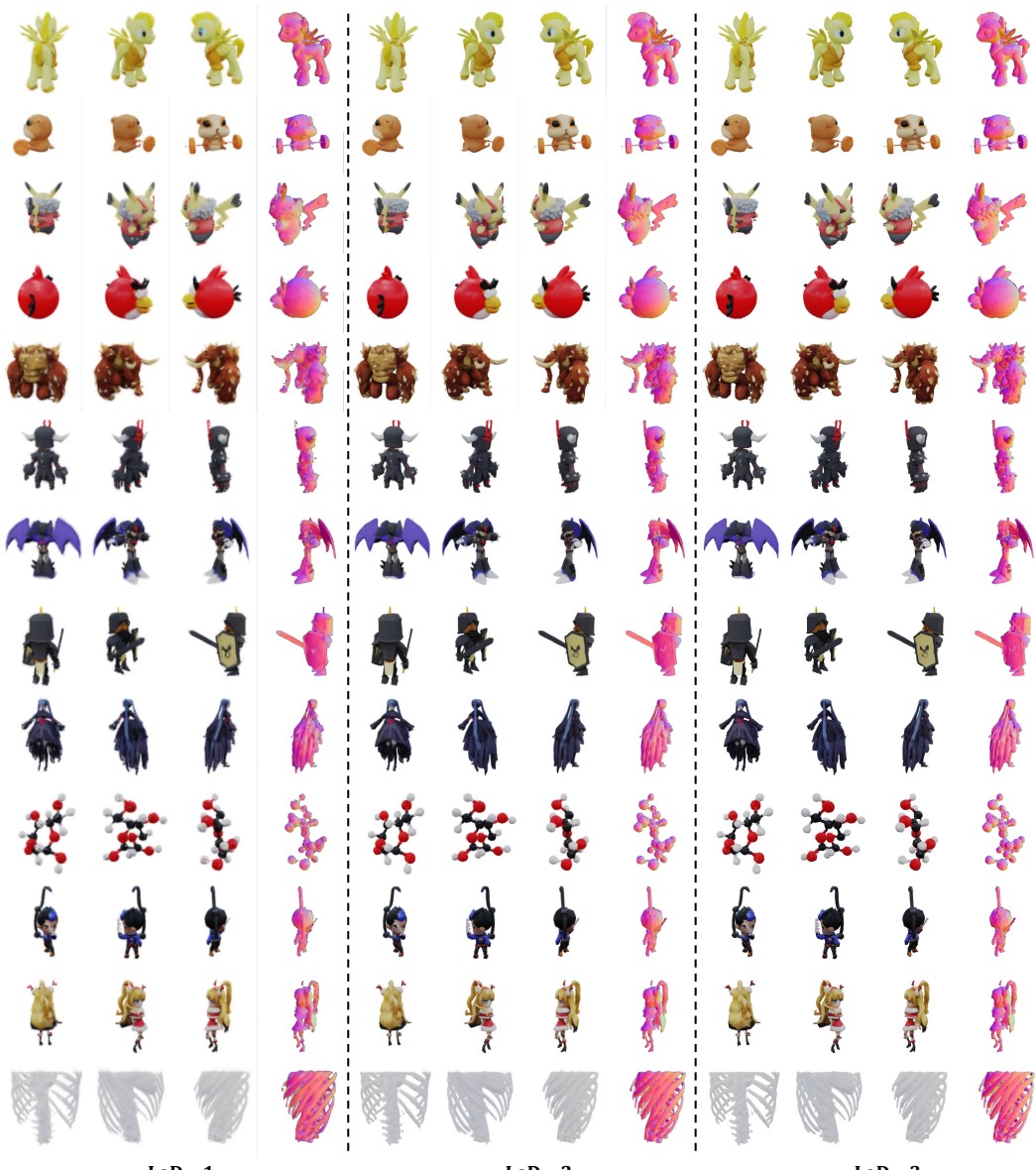

LoD = 1          LoD = 2          LoD = 3

Figure 8: **3D VAE Reconstruction**. Here, we visualize the 3D VAE reconstruction performance across different level of details (LoD). As shown, higher LoD results in sharper textures and smoother surface. Better zoom in.

generation (Huang et al., 2024b) and mesh generation (Siddiqui et al., 2023; Chen et al., 2024c) for producing high-quality object-level surfel Gaussians.

**Broader Social Impact.** In this paper, we introduce a new latent 3D diffusion model designed to produce high-quality surfel Gaussians using a single model. As a result, our approach has the potential to be applied to generate DeepFakes or deceptive 3D assets, facilitating the creation of falsified images or videos. This raises concerns, as individuals could exploit such technology with malicious intent, aiming to spread misinformation or tarnish reputations.

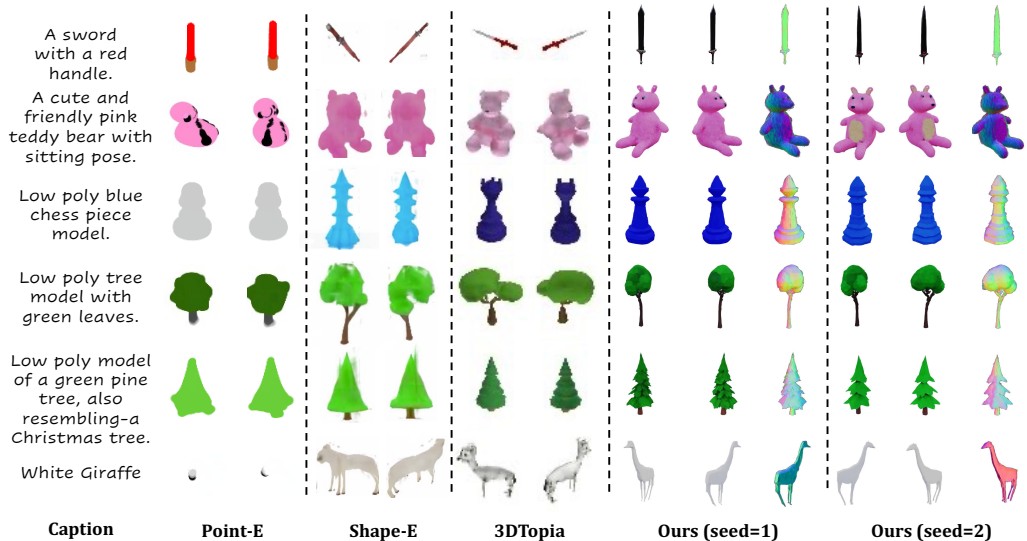

Figure 9: **More Qualitative Comparison of Text-to-3D**. We present more text-conditioned 3D objects generated by GAUSSIANANYTHING, alongside comparisons with competitive alternatives, including Point-E, Shape-E, and 3DTopia. As demonstrated, our approach consistently achieves superior quality in geometry, texture, and alignment between text and 3D content.

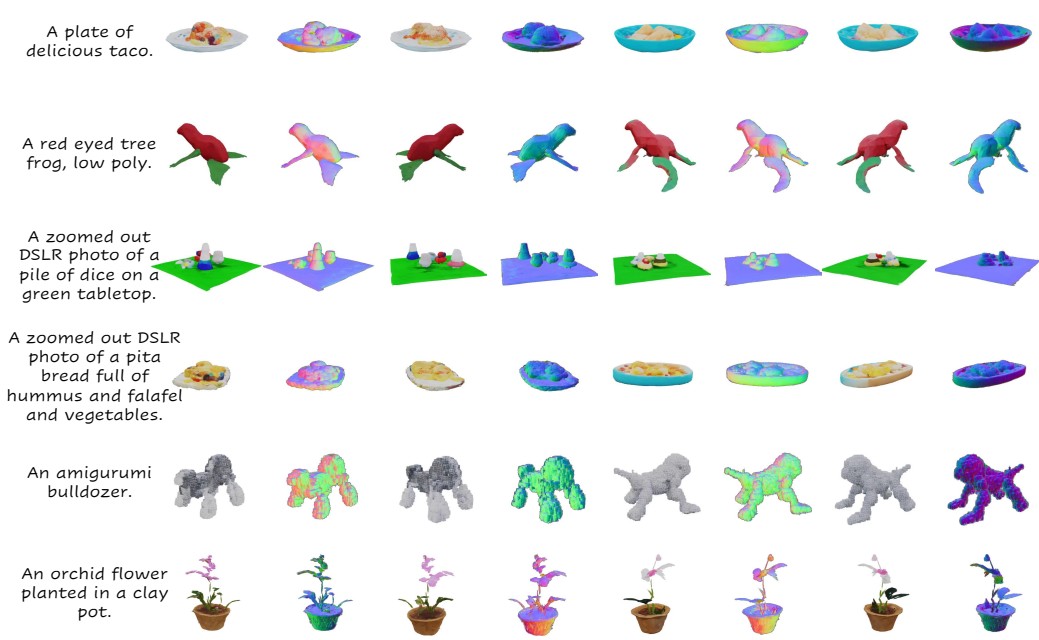

Figure 10: **More Qualitative Results of Text-to-3D over DF-415 Captions**. Our proposed method generalizes to long captions with detailed descriptions. All results are uncurated.

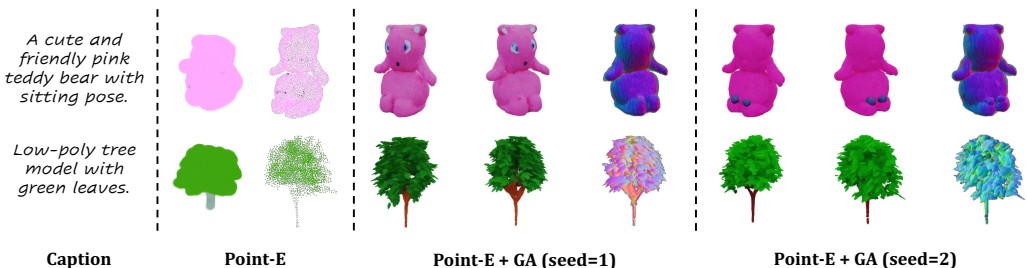

Figure 11: **Cascaded Text-to-3D with Point-E**. Thanks to our cascaded 3D generation design, the stage-2 diffusion model in GAUSSIANANYTHING seamlessly integrates with other point cloud generative models. To illustrate this capability, we leverage the state-of-the-art Point-E model. As demonstrated, our stage-2 diffusion model effectively transforms the point clouds generated by Point-E into diverse surfel Gaussians with more visually appealing features and enhanced geometric details.

