# OpenReview forum: "GaussianAnything: Interactive Point Cloud Flow Matching for 3D Generation"
_ICLR.cc/2025/Conference — ICLR 2025 Poster_

### Official Review · Reviewer_45wi · 2024-10-28

**Soundness:** 3
**Presentation:** 3
**Contribution:** 3
**Rating:** 6
**Confidence:** 4

**Summary:**

- This paper introduces a text- or image-to-3D generation framework using a VAE that processes multi-view RGB, normal, XYZ, and camera embeddings to construct a "Point-Cloud structured" Latent space, followed by a 3D DiT to decode 3D Gaussian parameters
- The proposed method employs cascaded geometric and texture latent diffusion models for effective shape-texture disentanglement, enhancing a "3D-aware editing" application.
- Experiments on text- and image-to-3D baselines consistently outperform existing methods.

**Strengths:**

- The paper proposes 3D-VAE and 3D-Diffusion models that are well-designed, flexibly supporting both Text- or Image-to-3D Generation and 3D Editing applications.
- Gaussian Anything improves "3D Gaussians Utilization", which reduces the rendering of white backgrounds, adaptively generates compact 3D Gaussians Splatting parameters and also benefits rendering quality.
- The paper is overall well-written, and the models (3D VAE and two-stage cascaded Diffusion model) are well-designed.
- It surpasses existing methods like 3DTopia, LN3Diff, and Shape-E in text-to-3D quantitative metrics and Lara/LGM in image-to-3D qualitative results.

**Weaknesses:**

- Qualitative metrics for Image-to-3D are **omitted**. Please add appropriate experiments in **Sec.** 5.1. Since the authors have rendered GSO datasets, GaussianAnything should be compared to LGM/Ln3Diff/Lara (refer to **Fig.** 3), and metrics such as PSNR, SSIM, LPIPS, or Image-CLIP Score may be reported.
- The claim that GaussionAnything can support point clouds or multi-view images as input lacks sufficient experimental proof (only VAE supports multi-view/point cloud as input). The authors should either abandon this claim or provide corresponding quantitative or qualitative results in **Sec.** 5.
- The use of flow matching for "velocity-prediction" in **Equations** 12 and 13, as opposed to direct "noise prediction", warrants further explanation in **Sec.** 4.2.


I would appreciate it if the authors could address my concerns by providing corresponding quantitative or qualitative results based on the **weaknesses** and **review feedback**.

**Questions:**

- Are there any insights during the proposed VAE training stage (see **Sec.** 4.1) regarding model design or loss functions?

---

> ### Author Response · Authors · 2024-11-21
> **Official rebuttal comments to Reviewer 45wi**
>
> **Qualitatively image-to-3D performance**. We have included the quantitative metrics for image-to-3D in Tab.2 as requested, which extensively evaluates both the rendering and geometry generation quality. Specifically, we report the CLIP-I, FID, KID, and MUSIQ metrics for visual quality evaluation, and Point Cloud FID (P-FID), Point Cloud KID (P-KID), Coverage Score (COV), and Minimum Matching Distance (MMD) for 3D quality metrics. As demonstrated, our method achieves state-of-the-art 3D generation performance over all the benchmarks, except for FID/KID metrics. Note that PSNR/LPIPS is not reported since different 3D generation methods have different output scale and canonical poses, which make it hard to align the predictions accurately. CLIP-I serves the same functionality here to evaluate the alignment between input and generated 3D.
>
> **Clarifications on the conditions**. GaussianAnything does not support multi-view images directly as input for now, and we have revised the claim in the abstract accordingly. For the point cloud condition, since our stage-2 diffusion model adopts both point cloud and image/text as the inputs, the point cloud condition is intrinsically supported by the GaussianAnything framework. Besides accepting generated/edited point cloud in stage-1, we demonstrate in Fig. 11 that our method also adopts point cloud generated from other pre-trained point cloud generative models, such as Point-E. This verifies the generalizability and flexibility of our proposed method.
>
> **Clarification on the flow matching objective.** “Velocity-prediction” is the default prediction setting in flow matching, as opposed to noise prediction in the DDPM training. In this paper, we also adopt the velocity prediction and follow the unified notations from SD3 in Eq. 6 and Eq. 7. Briefly speaking, the objective transforms the velocity-relevant terms into the weight part, so the conventional eps-pred diffusion loss is still used here. The detailed derivation is included in the appendix A.3, from equations (13)-(27).
>
> **Insights of 3D VAE training.** Regarding the 3D VAE training, some key designs help.
>
> 1. To maximize the benefits of the transformer's flexibility for 3D-aware operations, we incorporate it extensively into our design. Specifically, we utilize transformer-based architectures for multi-view attention in the VAE encoder and for 2D Gaussian Splatting (2DGS) up-sampling in the decoder. Additionally, adopting a pre-norm architecture effectively stabilizes training, ensuring more reliable convergence.
>
> 2. We also prioritize incorporating as many 3D attributes as possible into the input. As shown in Table 3, using a rich set of attributes, including RGB-D-N-Plücker coordinates, significantly enhances the quality of 3D encoding by providing comprehensive geometric and photometric information.
>
> 3. For 3D latent prediction, we use ground-truth sampled point clouds as anchor points. During our experiments, we found it challenging to regress z_0​ directly from multi-view inputs, likely due to the difficulty of spatial compression for point clouds, which demands specialized operations. To address this, we use the XYZ coordinates sampled directly from the 3D object surface as z_0​, augmented with learnable z_x​, as illustrated in Fig. 1. This approach facilitates accurate latent prediction while leveraging the spatial structure of the point cloud.
>
> 4. Finally, in the 2DGS up-sampling stage, we predict residuals instead of directly generating final outputs. This residual-based approach improves the quality of predictions by focusing on refining the details of the generated results.

---

> > ### Comment · Reviewer_45wi · 2024-11-22
> > **Official Comment by Reviewer 45wi**
> >
> > I sincerely appreciate the authors' response.
> > - I acknowledge that substantial engineering efforts are required to align the object scale and canonical pose settings for numerous 3D generation methods. Among the proposed metrics (Tab. 2), GaussianAnything exhibits superior performance compared to other approaches, except for FID and KID.
> > - It is compelling to observe that the stage-2 diffusion model can handle point clouds generated by Point-E. However, claims of X-anything's versatility should be cautious, as it may struggle to handle real-world point clouds from diverse sources.
> > - I appreciate the clarification on the flow matching objective and 3D VAE training, which resolves most of my concerns.
> >
> > I find the concept of GaussianAnything is interesting in the realm of native 3D diffusion. Nonetheless, the texture results are somewhat unsatisfactory and the text-to-3D results indeed appear to be rudimentary, indicating room for improvement. Considering the overall aspects, I will keep my score.

---

> > > ### Author Response · Authors · 2024-11-23
> > > **Official rebuttal comments to Reviewer 45wi**
> > >
> > > First, thanks for your timely response we really appreciate your insightful suggestions to our work!
> > >
> > > 1. We have revised our writing in L:1079 and cautiously claim that GaussianAnything supports point cloud-conditioned 3D generation only at the object level.
> > >
> > > 2. Our method is currently trained over limited resources and 3D datasets, and we will further scale up our training and further improve our performance regarding the texture qualities over both image and text-based 3D generation.
> > >
> > > Thanks again for your appreciation of our work.

---

### Official Review · Reviewer_21hH · 2024-10-30

**Soundness:** 3
**Presentation:** 4
**Contribution:** 3
**Rating:** 6
**Confidence:** 3

**Summary:**

This paper focuses on feed-forward 3D generation, introducing an elegant 3D latent diffusion framework.

The process begins by training a point cloud-structured latent representation—unlike the tri-plane representation commonly used in LRM, 3DTopia, and LN3Diff—through a VAE, using multi-view images, depths, and normals as input. The decoder of the VAE consists of a DiT transformer to predict coarse Gaussians, and a few cascaded upsamplers to predict dense Gaussians. The VAE is trained with a combination of a rendering loss, geometry regularization, KL constraints as well as an adversarial loss.
Next, a cascaded latent diffusion model is trained using flow matching for disentangled geometry and texture generation, which enables interative 3D editing.
The proposed framework can handle various conditioning inputs, including point clouds, texts and images.

**Strengths:**

The paper is clearly presented, with a well-motivated and thoughtfully designed method. I appreciate the overall approach and its intuitive structure.

**Weaknesses:**

The generated 3D objects presented in the paper and video demos are relatively simple and lack high quality, which diminishes the overall effectiveness of the method.
There are no quantitative results on image-to-3D generation.

**Questions:**

What dataset is used for text-to-3D evaluation? It seems not the commonly used T^3 benchmark. Why are metrics like render-FID used in CLAY (Zhang etal. 2024) not used here?

Can the method generate 3D objects with more complex geometries and higher quality?

Can PBR modeling be incorporated into the proposed framework?

As mentioned in the paper, the intermediate Gaussian output naturally serves as K LoD, allowing for a balance between rendering speed and quality in various scenarios.  I would be interested in seeing visualizations of the Gaussians at different LoD learned by the decoder. Additionally, how long does it take to perform one upsampling?

I would also like to see a discussion on the limitations.

---

> ### Author Response · Authors · 2024-11-21
> **Official rebuttal comments to Reviewer 21hH**
>
> **Evaluations on the 3D generation.** As explained in R1, we adopt the same text evaluation pipeline of 3DTopia. For text-to-3D evaluation, we adopt the CLIP score as the common metric and include the CLIP-I, Render-FID, and Point-FID for the image-to-3D evaluation.
>
> **More T23D results.** We have included more text-to-3D results in Fig. 7, Fig. 9, Fig. 10, and Fig. 11. In Fig. 9 and Fig. 10, we especially verify our method’s capability over complex long captions selected over the DF-415 dataset. To further boost the generation performance, using larger and more diverse 3D datasets like Objaverse-XL and MVImageNet will be helpful. We will explore these designs in the future.
>
> **Support of PBR.** Our method can directly support PBR by changing the final 2DGS rendering techniques to PBR-based Gaussian rendering techniques, such as [3D Gaussian Ray Tracing: Fast Tracing of Particle Scenes, TOG 2024]. This can be achieved by fine-tuning the pre-trained 3D VAE decoder only with the PBR rendering pipeline.
>
> **LoD visualization.** We have included the K LoD VAE reconstruction results in Fig. 8 in the appendix. Specifically, 3 LoDs are shown here and reflect the trade-off between rendering speed and quality. Each upsampling is achieved by a two-layer transformer block and requires 4.921, 9.518, and 9.068 milliseconds for the three stages up-sampling correspondingly. The speed is profiled on the A100 GPU with batch size 1.
>
> **Discussions on the limitations**:
>
> *1. Enhancing the 3D VAE Quality*: The performance of the 3D VAE could be improved by increasing the number of latent points and incorporating a pixel-aligned 3D reconstruction paradigm, such as PiFU (ICCV 2019), to achieve finer-grained geometry and texture alignment.
>
> *2. Incorporating Additional Losses in Diffusion Training*: Currently, the diffusion training relies solely on latent-space flow matching. Prior work, such as DMV3D (ICLR 2024), demonstrates that incorporating a rendering loss can significantly enhance the synthesis of high-quality 3D textures. Adding reconstruction supervision during diffusion training is another promising avenue to improve output fidelity.
> *3. Leveraging 2D Pre-training Priors*: At present, the models are trained exclusively on 3D datasets and do not utilize 2D pre-training priors as effectively as multi-view (MV)-based 3D generative models. A potential improvement is to incorporate 2D priors more effectively, for instance, by using multi-view synthesized images as conditioning during training instead of single-view images.
>
> *4. Expanding Dataset Diversity*: Utilizing more diverse and extensive 3D datasets, such as Objaverse-XL and MVImageNet, could further enhance the quality and generalizability of 3D generation.
>
> By addressing these aspects, the proposed method could achieve significant advancements in both the quality and versatility of its 3D generation capabilities.

---

> > ### Comment · Reviewer_21hH · 2024-11-26
> >
> > I appreciate the authors' responses. All my concerns are well addressed. I will keep my score.

---

> > > ### Author Response · Authors · 2024-11-27
> > > **Reply to Reviewer 21hH**
> > >
> > > Dear Reviewer 21hH,
> > >
> > > Thanks for your timely response we really appreciate your insightful suggestions to our work! We will include all the suggestions in the final version to improve the soundness of our submission.

---

### Official Review · Reviewer_PmZF · 2024-11-03

**Soundness:** 3
**Presentation:** 3
**Contribution:** 3
**Rating:** 6
**Confidence:** 2

**Summary:**

The paper proposes a unified 3D generation framework using a flow-matching network to sample pre-trained point cloud structured latent in a cascaded manner to achieve geometry-texture disentanglement. The paper uses multiview RGB-DN images as input to obtain point cloud structure latent vectors. These vectors are then used to generate Gaussian Surfels through a transformer decoder and further upsampled to obtain a dense Gaussian point cloud for rasterization. This stage is trained with a VAE objective to learn a latent distribution over the latent vectors. In the second stage, a flow-matching network is used to sample the point cloud structured latent vectors in a cascading manner. This achieves geometry and texture disentanglement and allows texture variation given sampled geometry. The proposed method generates visually impressive results compared to many baselines. It also contains a rich set of ablation studies.

**Strengths:**

1. The paper shows better generation results compared to competitive baselines such as GRM. The qualitative visuals also look promising in, for example, generating the thin structures in the first row of Fig. (3).
2. The ablation study is comprehensive. It shows that the authors clearly put lots of efforts into validating the proposed system.
3. The proposed system enables many applications including editing and generation tasks. This is further strengthened by its ability to disentangle geometry and texture.

**Weaknesses:**

1. Qualitative examples. In Fig. (3) why most of the input images are very dark? It seems that perhaps the other baselines wouldn't do well because of the particular input image choice. Also, the dark image makes it hard for me to judge if the proposed method is actually better than the baselines. For example, it's hard to see the color of the cactus on row 2. Also, why do all the baselines have different poses from each other? This also makes comparison difficult.
2. For Fig. (6b), why editing on latent space better than directly editing on Gaussians based on this example? Both show disjoint geometry.
3. The generation results shown in the paper do not look very complex compared to the visuals in existing works such as GRM. It would be great if authors could provide further visual comparisons on more diverse prompt and caption inputs.

**Questions:**

1. Eq. (5) misses a parenthesis.
2. The sentence from Ln 49 - 51 is difficult to understand. What do the authors mean by the set as the latent space?
3.  Ln. 52-54. "Specifically, we project the un-structured features $\mathbf{z}_z$ onto the manifold of the input 3D shape through the cross attention layer:" The word "manifold" here is unnecessary and can lead to confusion since the queries are just fixed resolution sampled points, not the actual shape.

---

> ### Author Response · Authors · 2024-11-21
> **Official rebuttal comments to Reviewer PmZF**
>
> **1. Qualitative evaluation issue**
>
> *Evaluation image issue.* As explained in A.2, we adopt the GSO renderings from the official Free3D repo for qualitative evaluation. After careful checking, we observe that their rendering adopts a single point lighting and some viewpoints of the object are constantly dim. We replaced those images in the updated Fig. 3 with bright and clear in-the-wild images for visual comparisons. More visual results are also included in Fig. 7.
>
> *Viewpoint issue.* The baselines have different poses since they have different canonical poses. Also, for LRM-like work, their canonical pose is dynamically determined by the input view. Therefore, objects generated from different baselines have different results even when rendered from the same given pose. We alleviate this issue by manually aligning the poses in the updated Fig. 3, with the video comparison in the supplementary (371-supplementary/rebuttal/371-i23d-video.mp4).
>
> **2. Editing issue.** We mean to disjoint the two parts here to demonstrate that editing on the latent code-based 3D editing yields smoother and more reasonable results, while 3D-space direct editing shows tearing artifacts. Off-the-shelf 2D editing techniques like self-guidance (Epstein et al, NeuIPS 2023) can be applied to achieve more versatile 3D editing (e.g., 3D drag). We leave these explorations as the future work.
>
> **3. More visual results.** More visual comparisons on image-to-3D are included in Fig. 3 and Fig. 7, and generations with more complex captions are included in Fig. 9, Fig. 10, and Fig. 11.
>
> **4. Writing issues** We have revised the parenthesis missing issue in the updated version.
> In Ln 249-251, we use the term set to denote un-ordered 3D latent space with no explicit 3D structure.  This terminology is aligned with “3DShape2VecSet”, and we will revise the writing accordingly.
> In Ln 252-254, we have revised the writing of “manifold” to “the sparse 3D point cloud” for accuracy.

---

> > ### Comment · Reviewer_PmZF · 2024-11-26
> >
> > Thanks for the authors' effort and time spent on the revised version. I appreciate the additional visual examples showcased in the paper and the supplementary and explanation of the issues I raised above. While the additional visuals further validate the methods' qualitative results, they are comparable qualitative-wise to the examples showcased in the baseline papers. Thus I will keep my original score.

---

> > > ### Author Response · Authors · 2024-11-26
> > > **Official reply to Reviewer PmZF**
> > >
> > > Dear reviewer PmZF,
> > >
> > > Thanks again for your insightful suggestions to our method. We will include all the discussions in the final revised version and further improve the quality of our proposed method as suggested.

---

### Official Review · Reviewer_cogY · 2024-11-04

**Soundness:** 3
**Presentation:** 3
**Contribution:** 2
**Rating:** 6
**Confidence:** 4

**Summary:**

This manuscript introduces a point cloud-based latent diffusion model for 3D Gaussian splatting generation. The proposed pipeline consists of a point cloud VAE and a cascaded diffusion model. The VAE encodes multiview RGBDN into a downsampled features, and employs cross attention between points sampled from the 3D surface and the features to aggregate the information into the points, yielding the featured points as the set latent. A transformer decoder is adopted to upsample the latents into 2DGS representation. The DiT based diffusion model generates first the position of the points, and then the features conditioned on the positions. Experiments have demonstrated its performance on text/image-to-3D generation and 3D editing.

**Strengths:**

- First of all, I think it's quite impressive to me that a 3D-native diffusion model can be trained from scratch to achieve the presented results, although my knowledge may be outdated since I'm not an expert in point cloud-based 3D generation. Many SOTAs (e.g. 3DTopia) gain advantage using 2D diffusion priors from large pretrained models, so I'm impressed that the proposed model can compete against some of those method.

- Great writing overall. The network architectures are very clearly presented.

- Geometry/texture disentanglement and the ability to edit the results in 3D add to the versatility of the proposed approach.

**Weaknesses:**

- I would disagree with the statements in L244-249. Why would a latent of shape $V\times (H/f)\times (W/f)\times C$ be difficult to generate for diffusion models? Isn't this what most multi-view image diffusion models are doing? My understanding is that, such high dimensional latents have more capacity and requires a larger model and a larger dataset to unleash its potential. Choosing a more compressed latent design is reasonable for this work since the model is trained from scratch, but it may not be the optimal choice when scaling up.

- Experiments are underwhelming in general, both quantitatively and qualitatively.
  - Quantitatively, the only comparisons made are on text-to-3D with CLIP being the only metric. The details of the experiment setup, such as the number of objects tested and the source of the captions, are not specified at all. Image-to-3D and editing abilities are not evaluated quantitatively, so the real potential of its versatility is not well supported.
  - Qualitatively, the few text-to-3D examples shown are not enough to demonstrate its generalization. The caption also appears very simple, unlike many previous approaches that adopt the more complex DreamFusion captions. Image-to-3D qualitative results are not satisfactory, given the missing divider of the first file sorter object and the slightly distorted shape of the final tower object. The competitors are also relatively weak. Many previous works (e.g. One2345++) can handle these simple objects easily. While it's understandable that many works are not open source, making fair comparison difficult, I find the performance of the proposed approach underwhelming in general.

- Using Dino features for image conditioning may introduce a major limitation for image-to-3D generation due to its large patch size and thus reduced resolution. Moreover, exact image-3D alignment gets more challenging in this case since the set latent doesn't really spatially align with any view. For these reasons, I'm doubtful about whether the model could produce faithful 3D results to the image. The first example in
Fig. 3 shows that the generated file sorter has 4 dividers while the input has 5, indicating the flaw in image-to-3D generation.

- The conditioning mechanism is not really unique at all. Any diffusion model can use modulated normalization and cross attention for text/image conditioning. In L363, the claim that the LRM line of work can only handle image condition is definitely wrong (Instant3D for text-to-3D).

**Questions:**

- I find L183 confusing. Flow matching models typically adopt the velocity parameterization $v=\epsilon - x_0$. Is the network predicting $\epsilon$ or $v$?

---

> ### Author Response · Authors · 2024-11-21
> **Official rebuttal comments to Reviewer cogY**
>
> **Statement of diffusion model training**  We have updated the original statement to ensure greater accuracy and clarity. While we acknowledge that the original statement was inaccurate, the advantages of our proposed latent space remain valid due to the following reasons:
>
> *1) Native 3D Diffusion*: Multi-view image diffusion models are not true 3D diffusion models and often face view inconsistency. In contrast, our point cloud-structured 3D latent space enables training native 3D diffusion models, addressing this limitation directly.
>
> *2) Scalability: Compressed latent spaces do not hinder scalability.* Models like Stable Diffusion and DiT demonstrate excellent scalability by training in compressed spaces. Similarly, our representation scales by increasing point clouds, enabling higher-dimensional latents with greater capacity, benefiting from larger models and datasets.
>
> *3) Improved Geometry*: Tab. 2 shows our method consistently outperforms multi-view reconstruction approaches like One-2-3-45. This highlights our approach’s suitability for 3D diffusion training, achieving superior geometry reconstruction at lower training costs.
>
> **Experiments comparison**
>
> *Experiment setup*. For the quantitative text-to-3D evaluation, we follow 3DTopia and use the prompt list proposed by “Gpt-4V is a human-aligned evaluator for text-to-3d generation” (CVPR 2024), which comprises 100 prompts generated by GPT-4.
>
> *Evaluation metrics*. We include CLIP-I, FID, KID, and MUSIQ for rendering metrics and  Point Cloud FID (P-FID), Point Cloud KID (P-KID), Coverage Score (COV), and Minimum Matching Distance (MMD) for 3D quality metrics.
>
> *Image-to-3D*. We include extensive evaluation on the image-conditioned 3D generation performance based on the above metrics in Tab. 2. As can be seen, our proposed method achieves the best performance except for FID (second-best). The evaluation details are also updated in the paper.
>
> *More text-to-3D examples with complex captions*. We present additional text-to-3D examples in Figures 7, 9, 10, and 11, using captions of varying complexity to showcase our method’s generalization and robustness. For fair comparison, most captions follow 3DTopia's style, with Figure 10 featuring more complex DreamFusion-inspired captions. Unlike optimization-based methods like DreamFusion, which leverage 2D diffusion priors trained on complex captions, our native 3D diffusion model is trained from scratch on Cap3D. Despite this, our method delivers comparable or superior results. While improved 3D caption annotations could enhance performance, this is a data-related issue beyond the scope of this paper.
>
> *Qualitatively image-to-3D performance*. We compare against strong competitors like Lara, which uses One2345++ for stage-1 multi-view generation, and demonstrate superior 3D reconstruction. Additional qualitative comparisons on challenging in-the-wild data are provided in Figures 3 and 7, with video renderings in the supplementary. Unlike Lara, which struggles with view consistency, our native 3D diffusion model enables stable, end-to-end 3D generation with intact and faithful reconstructions.
>
> **DINO features and conditioning mechanism**
>
> *DINO features*. Leveraging DINO features for image conditioning is standard practice in 3D generation, as demonstrated by recent works like Instant3D (ICLR '24), LRM (ICLR '24), CLAY (TOG '24), and LGM (ECCV '24). Besides, we use the DINO ViT-L network with a 518×518 resolution input and 14×14 patch size, and encode the input image into 1369 tokens. This ensures efficient processing without creating a bottleneck in the 3D generation pipeline.
>
> *Image-3D alignment*. As shown in Fig. 3, our method surpasses pixel-aligned approaches in novel views, thanks to our native 3D diffusion model design. Supplementary video renderings (371-supplementary/rebuttal/371-i23d-video.mp4) also highlight our method’s superior faithfulness. While methods like Splatter Image perform well over input views, they produce blurry or collapsed novel views. The CLIP-I metric in Table 2 further confirms our method’s superior consistency.
>
> *Conditioning mechanism*. We argue that we did not claim the CrossAttention-based conditioning as a novelty or a contribution in our paper. We just adopt this modern network design and present it in the paper for clarity.
>
> *LRM*. The LRM is a deterministic regression model which is designed to take image(s) as the input. To support text-conditioned 3D generation, a high-quality text-to-multiview model like Instant3D is required as preprocessing. However, in this cascaded framework, the 3D generation is intrinsically handled by the Instant3D side, not the LRM side. We have clarified the writing accordingly.
>
> **Questions regarding Flow Matching formulation** We follow the velocity parameterization of flow matching and directly predict v. For detailed derivation of our formulation in Eq. (6) and Eq. (7), please refer to Sec. A.3 in the appendix, and Eqs. (13)-(27).

---

> > ### Comment · Reviewer_cogY · 2024-11-23
> >
> > Thank you for the response! This has cleared many of my concerns.
> >
> > Regarding image-to-3D. I appreciate the addition of Table 2 and Figure 3. These results make the position of the proposed method very clear -- it may not be the best solution compared to multi-view methods, but it does show strong performance and outperforms other 3D-native models.
> >
> > > **Evaluation metrics. We include CLIP-I, FID, KID, and MUSIQ for rendering metrics and Point Cloud FID (P-FID), Point Cloud KID (P-KID), Coverage Score (COV), and Minimum Matching Distance (MMD) for 3D quality metrics.**
> >
> > I still don't see other metrics for text-to-3D. Is it missing in the revised PDF? I recommend adding the Aesthetic score metric, which seems to align well with texture details according to *Gpt-4V is a human-aligned evaluator for text-to-3d generation*.
> >
> > Fig. 10 doesn't appear to have actual comparisons with other methods despite the caption saying so.
> >
> > Still, the visual quality of text-to-3D is a bit under whelming since all the results are very simple objects. I'm not sure if this is the case for all 3D-native models, but adding the Aesthetic score metric to compare the texture details could make up for this.

---

> > > ### Author Response · Authors · 2024-11-27
> > > **Reply to Reviewer cogY**
> > >
> > > Dear Reviewer cogY,
> > >
> > > As the rebuttal phase approaches its end with only one day remaining for PDF revisions, we would like to kindly remind you of our responses to your comments and questions.
> > >
> > > If you have any remaining questions or concerns, we would be happy to discuss them further and make additional revisions as needed. Otherwise, if you find our updates satisfactory, we kindly invite you to consider reevaluating your score.
> > >
> > > Thank you again for your time, thoughtful feedback, and invaluable contributions to improving our work.

---

> ### Author Response · Authors · 2024-11-23
> **Official rebuttal comments to Reviewer cogY**
>
> Thanks for your timely response and insightful suggestions to our work.
>
> Regarding the evaluation metrics for Text-to-3D, originally we follow the previous methods (3DTopia, LN3Diff) and only reported CLIP score. As suggested, we have added two Aesthetic score metrics (MUSIQ-AVA [A] and Q-Align [B]) as mentioned in L:414 and updated Tab. 1 accordingly. As shown here, our method achieves better performance over the Aesthetic scores by a clear margin. Note that Q-Align [B] leveraging LLMs is the SOTA method for image aesthetic assessment.
>
> Regarding Fig. 10, we mean to include more visual results of our method over long captions and visualize the results over diverse viewpoints and random seeds. We have revised the captions as suggested.
>
> As also mentioned in our "rebuttal reply to Reviewer 21hH", the text-to-3D performance can be further improved by incorporating more diverse 3D datasets and scaling up the model. Currently all 3D-native generative models still fall behind optimization-based (e.g., SDS-based) 3D generation by a clear margin, but support speedy 3D generation within seconds. We believe our method has made a step forward to bridge the performance gap and can inspire future methods to further improve in this direction.
>
> [A] MUSIQ: Multi-scale Image Quality Transformer, Ke et al, ICCV 2021
>
> [B] Q-Align: Teaching LMMs for Visual Scoring via Discrete Text-Defined Levels, Wu et al, ICML 2024

---

> ### Comment · Reviewer_cogY · 2024-11-27
>
> Thank you for including the metrics in time!
>
> I will raise my score accordingly; however, I still find the overall generation quality somewhat lacking, as noted by other reviewers. While it is understandable that multi-view diffusion models perform better in this regard due to leveraging foundation image models, this paper does not adequately acknowledge this fact. For instance, the abstract claims that the method "outperforms existing methods in both text- and image-conditioned 3D generation," which is inaccurate. This is also not discussed in the limitations.
>
> Could you consider incorporating a stronger multi-view baseline, such as InstantMesh or GRM (GRM doesn't appear to have source code but there is an official demo, which provides api calls for inference)? It would also be helpful to explicitly clarify that there remains a performance gap between 3D-native models and fine-tuned multi-view models.

---

> > ### Author Response · Authors · 2024-11-27
> > **Reply to Reviewer cogY**
> >
> > We thank Reviewer cogY for the insightful suggestions.
> >
> > Regarding accessing the 3D generation quality, we believe there are two main manifolds: **3D geometry quality** and **rendering texture quality**. Geometry side, as denoted in Tab.2, our proposed method *quantitatively outperforms multi-view based 3D generation methods by a clear margin*. This phenomenon is also demonstrated in Fig. 3 (e.g., the house in row-1) that multi-view based methods tend to yield distorted 3D geometry. Texture side, though multi-view alternatives yield visually appealing textures and outperforms in FID/KID metrics, they lag behind in the novel view rendering quality. Comparatively, our proposed method achieves a higher CLIP-Image score compared to existing alternatives, thanks to the consistent 3D generation performance across all rendering views. This observation can be verified in the demo video and the video comparison in the supplementary (371-supplementary/rebuttal/371-i23d-video.mp4).
> >
> > Regarding the discussion or limitations, we have revised the argument/discussion in the paper to reflect this clearly (L:1019-L1045), and also revised the claim in the abstract.
> >
> > As requested, we will include the comprehensive qualitative and quantitative comparisons against stronger multi-view baselines like InstantMesh or GRM afterwards. This will further clarify the current texture gap between 3D-native models against multi-view methods.

---

> > > ### Comment · Reviewer_cogY · 2024-11-27
> > >
> > > Thank you for the quick response! My concern is that, SOTA multi-view methods such as InstantMesh or GRM also appear to have very good geometry. For text-to-3D, for example, GRM can understand more complex text prompts and generate complex objects, not just complex texture but also complex geometry details. It is necessary to compare with these methods if the proposed method is claimed to outperform multi-view methods.

---

> ### Author Response · Authors · 2024-11-27
> **Official reply to Reviewer cogY**
>
> We thank Reviewer cogY for the prompt response.
>
> 1. First, we need to emphasize that our method focuses on **a native 3D generative model with a point cloud-structured latent space that enables direct 3D synthesis and editing, a capability not achieved with multi-view-based 3D reconstruction methods.** While both approaches have their merits, they represent distinct and valuable lines of research.
>
> 2. Although our primary goal is not to surpass multi-view methods in visual quality, our geometry evaluation (shown in Tab. 2) demonstrates that our method outperforms them regarding geometric consistency and robustness. Besides, our method achieves comparable in appearance metrics, excelling CLIP-I and MUSIQ scores.
>
> 3. We want to point out that multi-view methods often exhibit unstable performance and failure cases in real-world scenarios. This may not be observed in cherry-picked successful cases, but we do observe this in practice and shown in Fig.3. This is because the image-to-multiview stage often generates 3D inconsistent images and thus causes distortion. In contrast, our native 3D methods demonstrate more stable performance across various cases. While GRM's code has not been released, it has almost the same architecture as LGM, which we have already compared. We will include additional comparisons and analysis with stronger multiview baselines in the final version.

---

> > ### Comment · Reviewer_cogY · 2024-12-02
> >
> > Thank you for the response.
> >
> > I understand that 3D native models and multi-view models are two distinct line of research. However, the manuscript does imply that the proposed method outperforms all other models, which lacks evidence.
> >
> > As I mentioned above, many multi-view-based models also produce good geometries. GRM is very different from LGM and its quality is much better. The code isn't available but the official demo provides Python APIs for inference. InstantMesh is also a very good choice, if GRM cannot be evaluated.

---

> ### Author Response · Authors · 2024-12-03
> **Official Comment to Reviewer cogY**
>
> Dear Reviewer cogY,
>
> Thanks for your clarifications regarding the concerns.
>
> First, we have **updated our manuscript and revised the claim of "outperforming all other methods" and have also included discussion of the limitations compared to multi-view based methods**. Specifically, our method outperforms native 3D diffusion model and shows comparative texture quality compared to multi-view based 3D reconstruction methods with better geometry consistency.
>
> Second, though GRM outperforms LGM, it remains within the multi-view generation + pixel-aligned GS prediction paradigm. Since no open-source code is available, we have used InstantMesh as the proxy for state-of-the-art multi-view methods. As requested, we have provided a quantitative evaluation of InstantMesh below. As can be seen, while InstantMesh performs SoTA on the texture metric CLIP-I and FID, it still lags behind our method in the geometry performance.
>
> |             | CLIP-I   | FID | P-FID      | P-KID    | COV       | MMD       | Prior Used | GPU Used |
> |-------------|-----------|------------|----------|-----------|-----------|-----------|-----------|-----------|
> | LGM         | 87.99    | 19.93 | 40.17      | 19.45    | 50.83     | 22.06     | MVDream | 32 A100 |
> | InstantMesh | **89.59** | **19.64**  | 27.17|    10.02 |     56.33 |     23.10 | OpenLRM & Zero-123++ | 8 H100 |
> | Ours        | 89.06    | 24.21 | **8.72**   | **3.22** | **59.50** | **15.48** | **None** | **8 A100**
>
> Moreover, we have included the prior model and GPU resources used in the comparisons. As can be seen, our **native point cloud-structured 3D diffusion model** does not leverage any existing priors and use the minimum computation resources. Therefore, it is reasonable that our design is currently inferior to the baseline methods regarding the texture fidelity. In the future, we will scale up our model and incorporate strong pre-trained priors to further improve the quality.

---

### Author Response · Authors · 2024-11-26
**Official Comment by Authors**

Dear Reviewers,

As the rebuttal phase approaches its end with only two days remaining for PDF revisions, we would like to kindly remind you of our responses to your comments and questions.

We have carefully addressed all your feedback and have made thoughtful updates to the manuscript to address your concerns. We hope that our efforts meet your expectations and provide clarity on the points you raised.

If you have any remaining questions or concerns, we would be happy to discuss them further and make additional revisions as needed. Otherwise, if you find our updates satisfactory, we kindly invite you to consider reevaluating your score.

Thank you again for your time, thoughtful feedback, and invaluable contributions to improving our work.

---

### Meta-Review · Area_Chair_HNCH · 2024-12-17

**Metareview:**

This paper introduces a novel 3D generative model using a VAE and a flow-matching model. The key idea is to use a latent point cloud representation that effectively disentangles geometry and texture information. The authors propose a cascaded generative model based on flow matching: first generating the point cloud, and then generating features for each point.

All reviewers agreed to accept the submission to ICLR 2025 after the rebuttal. The reviewers noted that the experiments added in the rebuttal addressed concerns about the comparison with previous work. However, some reviewers still raised concerns about the unsatisfactory quality of the qualitative results and the inconsistency with the input images. Despite these issues, the reviewers agreed to accept the submission due to the additional quantitative results on geometry. Nevertheless, it is strongly recommended that the qualitative results be improved for the final version.

As a minor comment: the authors interchangeably use the terms diffusion models and flow-based models, even though these terms are distinguished in many contexts. The AC strongly recommends replacing the term diffusion models with flow-based models consistently throughout the entire paper.

**Additional Comments On Reviewer Discussion:**

Please see the Metareview.

---

### Decision · Program_Chairs · 2025-01-22

Accept (Poster)